# Deep level transient spectroscopic investigation of phosphorus-doped silicon by self-assembled molecular monolayers

Xuejiao Gao [1], Bin Guan [1], Abdelmadjid Mesli[2], Kaixiang Chen[1] & Yaping Dan [1]

It is known that self-assembled molecular monolayer doping technique has the advantages of forming ultra-shallow junctions and introducing minimal defects in semiconductors. In this paper, we report however the formation of carbon-related defects in the molecular monolayer-doped silicon as detected by deep-level transient spectroscopy and low-temperature Hall measurements. The molecular monolayer doping process is performed by modifying silicon substrate with phosphorus-containing molecules and annealing at high temperature. The subsequent rapid thermal annealing drives phosphorus dopants along with carbon contaminants into the silicon substrate, resulting in a dramatic decrease of sheet resistance for the intrinsic silicon substrate. Low-temperature Hall measurements and secondary ion mass spectrometry indicate that phosphorus is the only electrically active dopant after the molecular monolayer doping. However, during this process, at least 20% of the phosphorus dopants are electrically deactivated. The deep-level transient spectroscopy shows that carbon-related defects are responsible for such deactivation.

[1] University of Michigan–Shanghai Jiao Tong University Joint Institute Shanghai Jiao Tong University, 800 Dong Chuan Road, Shanghai, 200240, China. [2] Institut Matériaux Microélectronique Nanosciences de Provence, UMR 6242 CNRS, Université Aix-Marseille, 13397 Marseille Cedex 20, France. Xuejiao Gao and Bin Guan contributed equally to this work. Correspondence and requests for materials should be addressed to Y.D. (email: yaping.dan@sjtu.edu.cn)

Self-assembled molecular monolayer (SAMM) doping is a potential doping technique to tackle the challenges in the formation of sub-10-nm ultra-shallow junction[1] and has the advantage of facilitating mass production and applicability to semiconductors like Si, Ge, InAs, GaAs, etc.[2–5]. In this technique, dopant-carrying molecules are first covalently immobilized on the semiconductor surface via surface reactions. Due to surface self-limiting property, the areal dose of dopant molecules can be modulated by varying reaction temperature[6], reaction time[6], molecule size[7], and the composition of the molecules[8,9]. Subsequently, the dopants are driven into the semiconductor bulk and activated by thermal annealing. Unlike the technique of ion implantation, no lattice damage is found during the dopant-incorporation process[2,8,10]. In addition, this technique is suitable for doping in complex geometry structures, such as nanopillar arrays[4] or fins in fin-FETs[11].

During the thermal annealing process, other atoms in the molecular monolayer such as oxygen, hydrogen, and especially carbon[12] can be driven into silicon together with the desired doping element. These impurities are difficult to detect due to their atomic nature and low concentrations. It remains an issue whether these unintentional impurities form complex defects and how these defects affect the electrical properties of the substrate. Longo et al.[13] have suspected the possible influence of unintentional carbon contamination during the doping process and hence reported a SAMM doping method to minimize carbon incorporation by breaking chemical bonds and releasing carbon at lower temperature than that of annealing. However, no detail information was given in their study on why carbon ligand was removed before thermal annealing and how they affect the electrical properties of the substrate. Shimizu et al.[12] investigated the diffusion behavior of carbon and oxygen contaminants in phosphorus-doped Si substrates by time-of-flight secondary ion mass spectrometry (ToF-SIMS) and atom probe tomography (APT), finding that the contaminants were limited to the first atomic layer and could be easily removed. Puglisi and coworkers[14] believed that a surface layer where silicon intermixed with carbon from dopant-carrying molecules was present after SAMM doping. However, with a significant solubility in silicon and a diffusion coefficient larger than phosphorus[15], it is likely that carbon forms active defects, which would have significant influence on the electrical properties of the substrate. For example, interstitial carbon can bond with group V elements like substitutional phosphorus, arsenic, and antimony forming the pairs $C_i$–$P_s$, $C_i$–$As_s$, and $C_i$–$Sb_s$ with multiple deep energy levels[16] corresponding to several atomic configurations.

Deep-level transient spectroscopy (DLTS) is a very sensitive technique to study defects in bulk semiconductors, providing information on energy levels and concentrations of related defects[17]. Tremendous efforts have been made to acquire energy levels of impurities like carbon, oxygen, hydrogen, and their complex in silicon by using DLTS[18–20]. In this paper, we employ DLTS to investigate defects formed by impurities in SAMM-doped silicon. The molecular monolayer grafting and doping are characterized by X-ray photoelectron spectroscopy (XPS) and van der Pauw measurements, respectively. The total phosphorus concentration and the active fraction are determined by secondary ion mass spectrometry (SIMS) and low-temperature Hall measurements, respectively. The DLTS study shows that carbon-related defects are present in the SAMM-doped silicon, resulting in the electrical annihilation of phosphorus dopants due to bonding with interstitial carbon.

## Results

We fabricated phosphorus-functionalized silicon as outlined in Fig. 1. Briefly, a freshly prepared hydrogen-terminated silicon (surface 1) was passivated with 5-hexenyl acetate (molecule 1) in Ar atmosphere at 95 °C for 16 h, yielding a surface with acetate terminus (surface 2). Subsequently the acetate surface was reduced into a hydroxyl-terminated surface (surface 3) by lithium aluminum hydride (LiAlH$_4$) in tetrahydrofuron (THF) at 70 °C for 2 h. The hydroxyl groups on the surface were reacted with alkylphosphate (molecule 2) in the presence of activation agent dicyclohexylcarbodiimide (DCC), forming phosphate ester, thus rendering phosphorus-functionalized silicon (surface 4).

Each step of modification was characterized by XPS as shown in Fig. 2 and Supplementary Figure 1. High-resolution narrow scan of C 1s for surface 2 (Fig. 2a) reveals a broad peak at 285.0 eV (FWHM 1.4 eV) related to aliphatic carbon-bonded carbon (C–C) from 5-hexenyl acetate. The broad peak has a side shoulder at 286.6 eV (FWHM 1.6 eV) attributed to oxygen-bonded carbon (C–O) and carbon adjunct to carbonyl (C̲(C=O)). The small bump at 289.0 eV (FWHM 1.6 eV) is assigned to the carbon of carbonyl (C=O). The integral peak area ratio of C–C, C–O/C̲ (C=O), and C=O is 6:2:1 consistent with the stoichiometric ratio of 5-hexenyl acetate (5:2:1) immobilized on the surface. To provide a reference for surface 4 later, we also examined P 2s XPS spectrum of surface 2 ranging from 175 to 210 eV for phosphorus signal. As expected, no phosphorus was detected, except two broad peaks (Fig. 2b) due to silicon plasmon loss[21]. For surface 3, the C 1s scan (Fig. 2c) shows the same three carbon components

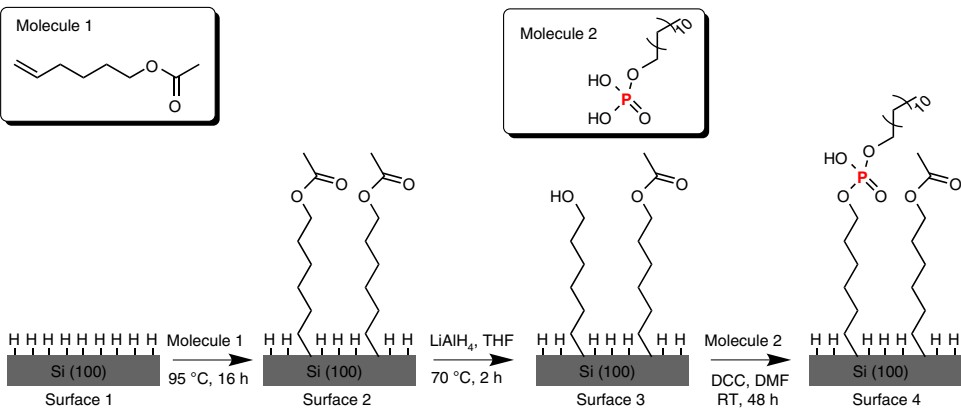

**Fig. 1** Stepwise surface modification on Si (100) surfaces. Molecule 1 is chemically grafted onto surface 1 under thermal treatment at 95 °C for 16 h forming a molecular monolayer on surface 2. Molecule 2 reacts with the hydroxyl group on surface 3 leading to a phosphorus-funtionalized surface 4

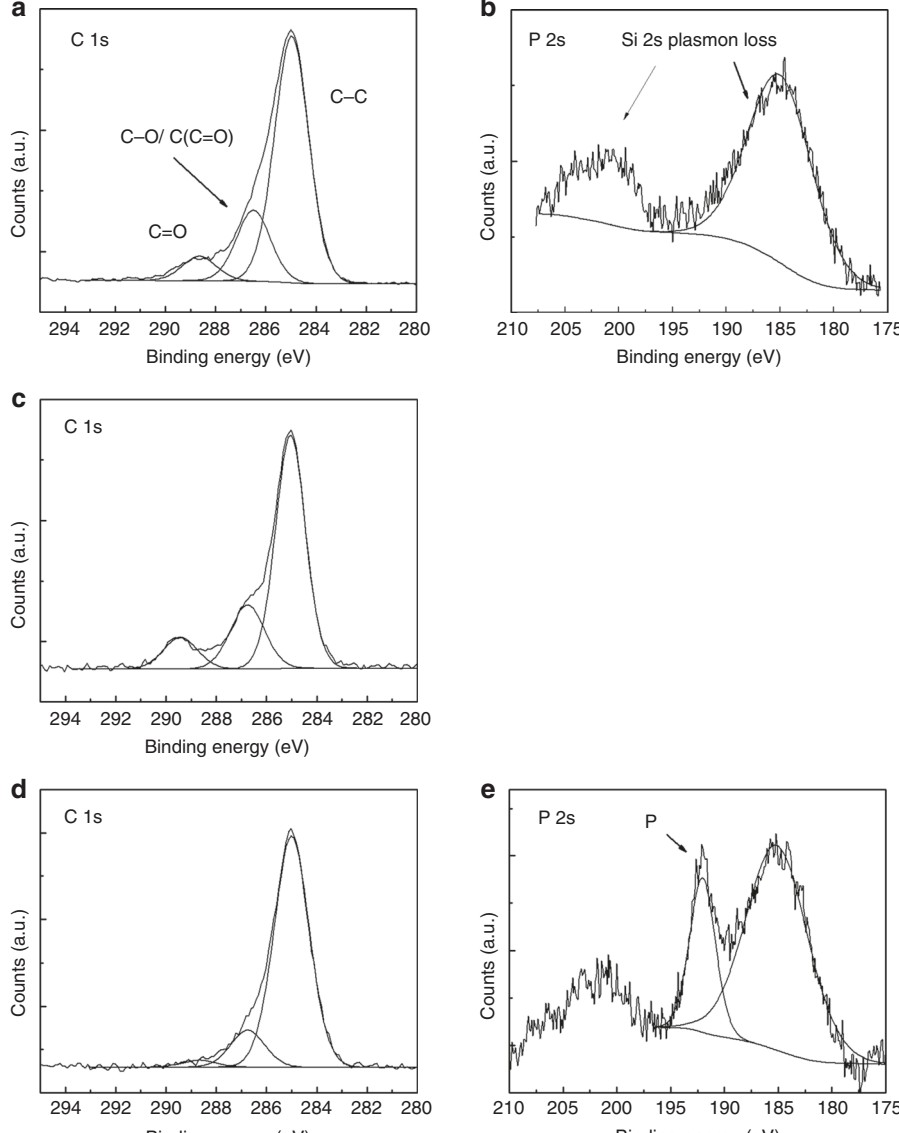

**Fig. 2** XPS spectra of modified silicon samples. **a** High-resolution narrow scans of C 1s and **b** P 2s obtained from 5-hexenyl acetate monolayers on silicon (surface 2 in Fig. 1). **c** C 1s spectrum of hydroxyl-terminated surface 3. **d** High-resolution scans of C 1s and **e** P 2s from phosphorus-modified silicon sample (surface 4)

| Table 1 Sheet resistances of silicon samples via SAMM doping technique by van der Pauw measurement | |
| --- | --- |
| **Si(100) intrinsic wafer, resistivity > 10 kΩ cm** | $R_s$ **(kΩ/sq)** |
| Unmodified sample (surface 1) | 317 |
| Control sample (annealed surface 3 with carbon monolayer) | 226 |
| Phosphorus-doped sample (annealed surface 4) | 1.06 |

as on surface 2, namely C–C, C–O/<u>C</u>(C=O), and C=O, with a peak area ratio of 10:3:1. This indicates that about half of acetate groups on the surface have been reduced to hydroxyl. For surface 4, the C 1s scan shows that the peak area ratio further increases to 30:5:1 (Fig. 2d), suggesting that the alkylphosphate is successfully coupled onto the Si surface. This successful coupling is also supported by an additional peak at 192.0 eV (FWHM 2.8 eV) in the P 2s spectrum (Fig. 2e) which is assigned to phosphorus from phosphate[22].

To drive the molecular-monolayer-carried phosphorus into the intrinsic silicon substrate (>10 kΩ cm), the chemically modified Si samples were first coated with $SiO_2$ made from spin-on-glass (SOG) and then annealed at 1050 °C for 2 min. The $SiO_2$ layer was later removed by buffered oxide etchant (BOE, HF:$NH_4$F = 6:1) before electrical characterizations. Van der Pauw four-point measurements[23] (Supplementary Note 1 and Supplementary Figure 2) were performed in darkness on the unmodified Si (surface 1), annealed surface 3 (as a control to phosphorus-doped sample), and surface 4 (phosphorus-doped sample). As shown in Table 1, the sheet resistance ($R_s$) for the control sample decreases slightly from 317 (for the undoped silicon) to 226 kΩ/sq, indicating no significant contamination introduced in the process. For the phosphorus-doped sample, the resistance drops dramatically to 1.06 kΩ/sq after doping. This suggests that the phosphorus dopants have diffused into and electrically doped the silicon substrate.

To examine the total amount of phosphorus incorporated into Si, the phosphorus-doped sample was analyzed by SIMS. As

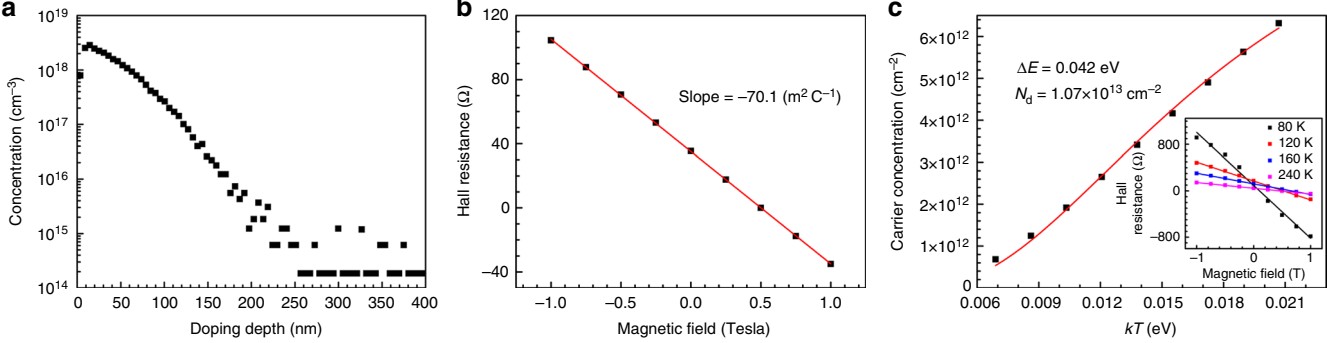

**Fig. 3** Dopant ionization rate. **a** Doping profile of phosphorus-doped Si measured by SIMS. **b** Hall resistance versus magnetic field measured by Hall measurement at room temperature. **c** Free electron concentration versus temperature. Inset: Hall measurements of phosphorus-doped Si at several temperatures

shown in Fig. 3a, the distribution of phosphorus dopants is highly non-uniform (see more discussions in Supplementary Note 2). The phosphorus concentration drops from around $3 \times 10^{18}$ cm$^{-3}$ by nearly three orders of magnitude within 200 nm below the surface. In terms of surface concentration, the phosphorus concentration per unit area is calculated to be $1.34 \times 10^{13}$ cm$^{-2}$ by integrating all phosphorus from the surface to bulk. To find out the free electron concentration of the phosphorus-doped samples, we performed Hall measurements. In Fig. 3b, the Hall resistance linearly changes with the applied magnetic field. The slope of the linear dependence is inversely proportional to the free electron concentration as shown in Eq. (1) from which the free electron concentration is found to be $8.92 \times 10^{12}$ cm$^{-2}$. Note that Eq. (1) is on the assumption of uniform doping. The non-uniform distribution of dopants in our sample may lead to a few percent errors in the obtained electron concentration (see Supplementary Note 3 for more discussions).

$$
\begin{aligned}
N_e &= -\frac{\Delta B}{e \times (\Delta V_H/I)} \\
&= -\frac{1}{e \times (\text{slope})} \\
&= \frac{1}{1.6 \times 10^{-19} \text{C} \times 70.1 \text{ m}^2 \text{ C}^{-1}} \\
&= 8.92 \times 10^{12} \text{ cm}^{-2}
\end{aligned}
\tag{1}
$$

in which $e$ is the unit charge, $V_H$ is the Hall voltage, $I$ is the source current, $B$ is the magnetic field, and $N_e$ is the free electron concentration per unit area.

Previously, it was reported that nitrogen carried by tert-butyl-N-allylcarbamate can electrically dope silicon[24]. To check whether other impurities besides phosphorus dopants are also electrically active in our doped sample, low-temperature Hall measurements were performed as shown in Fig. 3c. The temperature was set from 80 K gradually up to 300 K. The electron concentration per unit area was obtained from Hall measurements at each temperature (Supplementary Figure 3 and Supplementary Table 1). As the electron concentration as a function of temperature follows Eq. (2)[24], the activation energy of phosphorus dopants was found as 42 meV by fitting Eq. (2) to the experimental data, which is close to the known value (45 meV) of phosphorus ionization energy in silicon[25]. This finding indicates that there is no significant amount of electrically active impurities other than phosphorus donors in the SAMM-doped sample. From the fitting, we also attained the concentration of electrically active phosphorus dopants, which is $1.07 \times 10^{13}$ cm$^{-2}$. The free electrons in the doped sample are believed to originate from this part of phosphorus dopants. Thus, the ionization rate at room temperature is estimated to be 83.4% if we divide the electron concentration $(8.92 \times 10^{12}$ cm$^{-2})$ by the electrically active

phosphorus dopants $(1.07 \times 10^{13}$ cm$^{-2})$. This ionization rate is reasonable, considering that the ionization rate of phosphorus dopants in high concentration (about $10^{18}$ cm$^{-3}$ in particular) is as low as 80%[26,27]. Quantitatively, a theoretical ionization rate for electrically active phosphorus with the same distribution and concentration $(1.07 \times 10^{13}$ cm$^{-2})$ was calculated considering the effects of the incomplete ionization[26,27] and internal electric field. The resultant ionization rate is 81.3% (Supplementary Note 4 and Supplementary Figure 4), in good agreement with the experimental value. It means that this part $(1.07 \times 10^{13}$ cm$^{-2})$ of electrically active phosphorus fits the classical case for phosphorus donors in silicon. Note that the total phosphorus dopant concentration detected by SIMS is $1.34 \times 10^{13}$ cm$^{-2}$. The interesting question is what happened to the remaining 20% $(=(1.34 - 1.07)/1.34)$ of the phosphorus dopants $(0.27 \times 10^{13}$ cm$^{-2})$. We speculate that the remaining phosphorus dopants are electrically annihilated by carbon-related defects.

$$
n_c = \frac{-N_c + \sqrt{N_c^2 + 8N_c N_D \exp\left(\frac{\Delta E}{kT}\right)}}{4\exp\left(\frac{\Delta E}{kT}\right)}
\tag{2}
$$

where $N_c$ is the effective density of states function which is defined as $N_c \approx 2\left(\frac{2\pi m_n^* kT}{h^2}\right)^{\frac{3}{2}} = w(kT)^{\frac{3}{2}}$ with $w$ being the constant related to the band structure of the semiconductor, $N_D$ is the concentration of donors, and $\Delta E$ is the activation energy which is equal to $(E_c - E_d)$ with $E_c$ and $E_d$ being the conduction band edge and the donor energy level, respectively.

To verify this hypothesis, DLTS measurements were performed on SAMM-doped samples. DLTS requires a Schottky contact to be formed on top of the SAMM-doped surface (Fig. 4a). The depletion region of the Schottky junction will be readily extended into the substrate bulk if an intrinsic substrate is used. As a result, the information extracted from DLTS will be mostly originating from the bulk. However, the impurities and defects introduced by the SAMM doping are dominantly located near the surface. To detect possible defects in this region, we prepared a set of new samples on phosphorus-doped n-type Si (100) substrate with a resistivity of 1–3 Ω cm (phosphorus concentration 1 − 5 × 10$^{15}$ cm$^{-3}$; carbon concentration <5 × 10$^{16}$ cm$^{-3}$) to confine the Schottky depletion region near the surface. The same SAMM doping process as described previously (on SAMM-doped surface 4) was conducted on the n-type substrate. The successful doping of phosphorus into the substrate was confirmed by SIMS (Fig. 4c). To form Schottky contact, a 150-nm-thick Au electrode was evaporated directly on the SAMM-doped surface, which had been cleaned with Piranha solution and hydrofluoric acid. Al film was evaporated on the back side of the substrate that had been

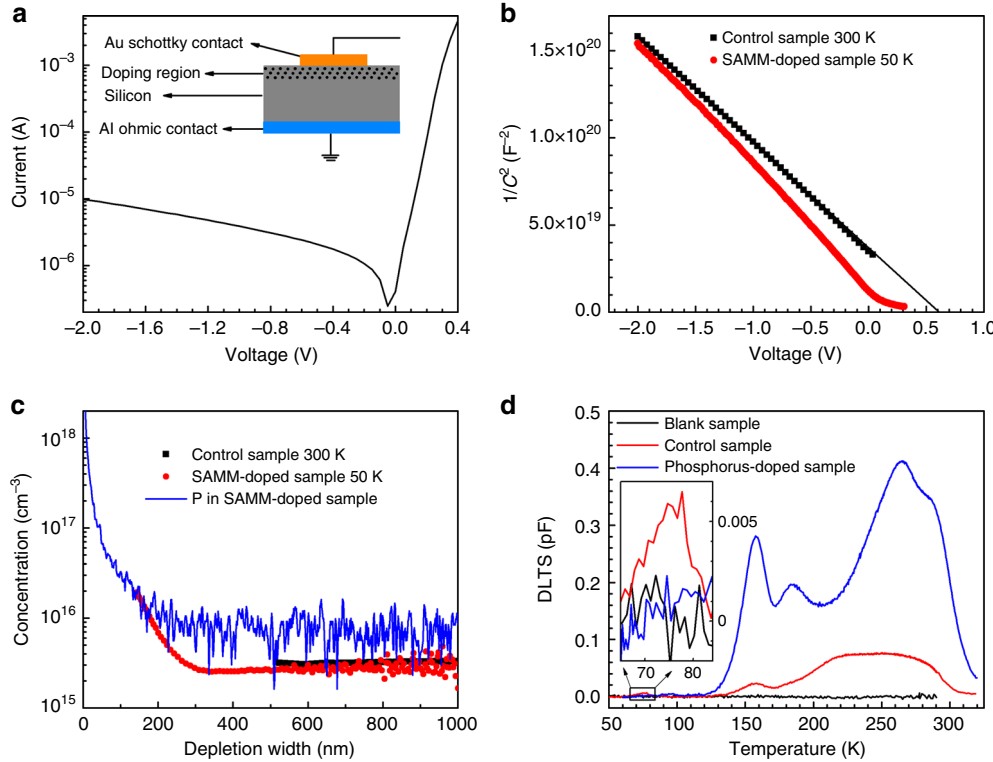

**Fig. 4** IV, CV, and DLTS data on SAMM-doped phosphorus-doped silicon. **a** *I–V* curve of the Schottky diode made on the SAMM-doped sample with the inset schematically showing the diode structure. **b** Capacitance as a function of bias voltage in form of $1/C^2$ versus *V*. **c** Charge carrier concentration at different depth derived from **b**. As a reference, phosphorus depth profile by SIMS is also presented in blue curve. **d** Comparison of DLTS spectra of the blank sample, control sample, and SAMM-doped sample with reversed-bias pulse from −2 to 0 V, at the rate window of 200 s$^{-1}$. The inset shows the spectra at the range of 65–85 K

extensively scratched by a diamond scribe. The scratch creates defects, which reduce the minority carrier lifetime and therefore facilitate the formation of Ohmic contact between the Al film and n-type silicon substrate (Supplementary Figure 5). No post-annealing was conducted to avoid Au/Al diffusion into silicon. The device schematic is shown in the inset of Fig. 4a. A typical *I–V* curve of the device is depicted in Fig. 4a, evidencing that a Schottky diode is formed. A similar process was also applied to the blank and control sample (both are n-type) to form Schottky contacts (Supplementary Figure 6) for DLTS measurements. The blank sample went through the SiO$_2$ capping and annealing process without any functionalization. The control sample went through all the processes except that the alkylphosphate was not added during esterification reaction, like surface 3 in Fig. 1.

The voltage-dependent capacitance of the Schottky junctions was first measured at 1 MHz with the dc bias sweeping from −2 to 0 V for the control sample and from −2 to 0.3 V for the SAMM-doped sample. Figure 4b shows the *C–V* dependence in form of $1/C^2$ versus dc voltage bias. For the control sample, the dependence is linear and the build-in potential is extracted as 0.57 V from the intercept with *x* coordinate. As expected, this built-in potential increases to 0.76 V as the temperature is lowered to 50 K (Supplementary Figure 7). For the SAMM-doped sample, the dependence of $1/C^2$ on dc voltage bias is nonlinear due to the highly non-uniform distribution of phosphorus dopants introduced by the SAMM doping process. This nonlinearity makes it unreliable to extract the built-in potential. But the ionized charge profile can be extracted, shown in red dots in Fig. 4c. The concentration of ionized charges in the control sample is around $3 \times 10^{15}$ cm$^{-3}$ (black dots in Fig. 4c) consistent with the nominal resistivity

$(1-3\,\Omega\,\text{cm})$ of the n-type Si substrate. In contrast, the ionized charge concentration in the SAMM-doped sample drops from about $2 \times 10^{16}$ cm$^{-3}$ at a depth of 140 nm to about $3 \times 10^{15}$ cm$^{-3}$ at about 330 nm, indicating that SAMM-introduced phosphorus diffuses beyond 300 nm. Note that the phosphorus concentration from SIMS (blue lines) is constant at about $10^{16}$ cm$^{-3}$ starting from a depth of 200 nm below the surface due to the relatively high detection limit of the SIMS technique.

DLTS measurements were performed on the samples at bias of −2 V with applied pulse of 0 V (hereafter it is written in form of "bias voltage"–"pulse voltage", i.e., −2 to 0 V) as shown in Fig. 4d. No peaks are detected for the blank sample (black curve), demonstrating that there is nearly no defects in bare silicon wafer and that the capping layer and the annealing process introduce no defects into silicon. For the carbon-chains-functionalized control sample (red curve), a tiny kink at 75 K (Fig. 4d inset) and a visible peak at 155 K next to a broad bump from 200 to 300 K are observed, indicating that carbon from the dopant-carrying molecules can diffuse into the substrate and produce some defects in phosphorus-doped Si. These defects could be related to C, H, O, and N. Oxygen plays a significant role only in the presence of lattice defects such as vacancies[28] which do not exist in the doping process considered in this work. Defects involving hydrogen are very unlikely as they do not exist after the high temperature treatments during which hydrogen out diffuses[29]. Finally, nitrogen if electrically active has very shallow energy levels, and thus none of the observed levels can be associated with this impurity, unless nitrogen binds to other unknown defects[30]. Therefore, we would attribute most of the observed defects to complexes where carbon is the main ingredient. For the sample doped by the molecular-monolayer-carried phosphorus (SAMM-

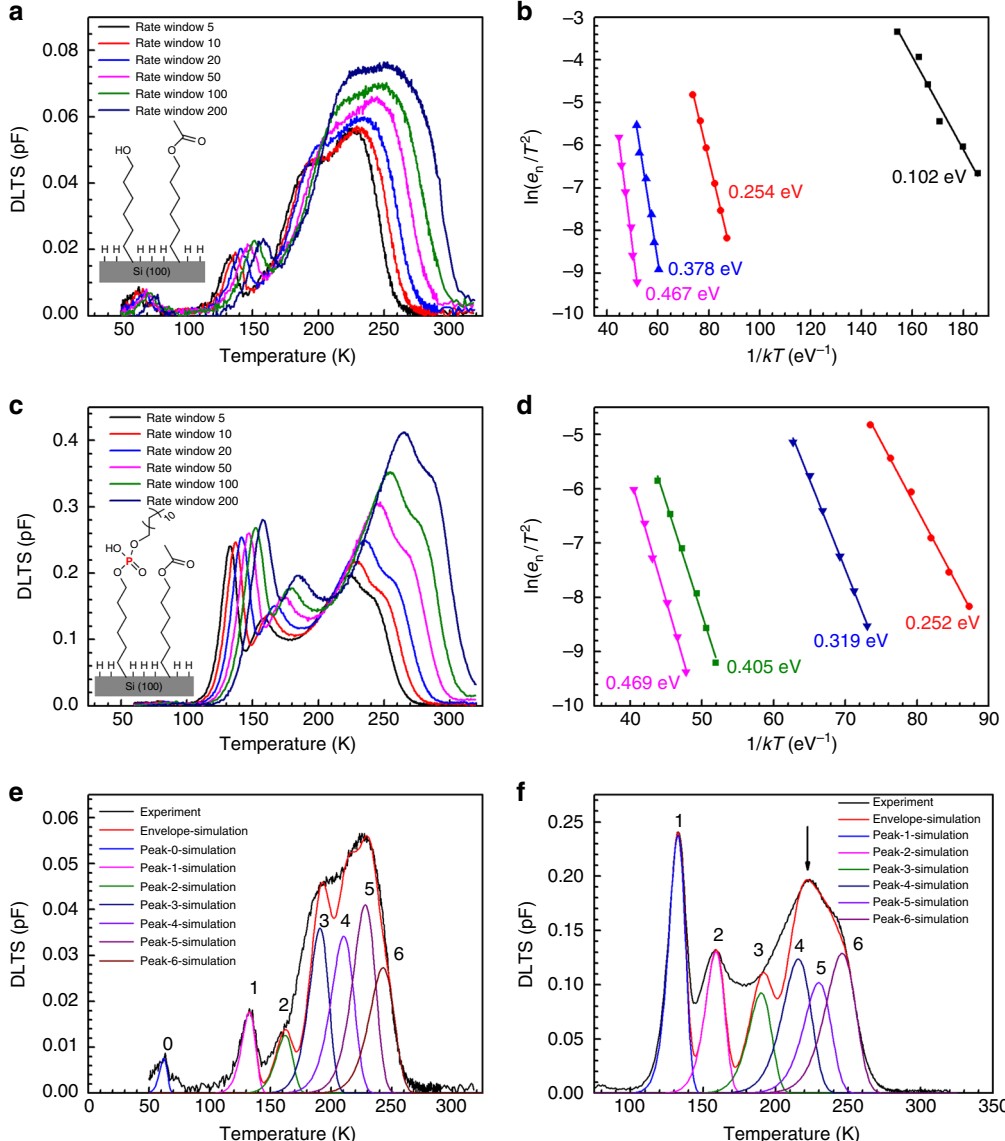

**Fig. 5** Defect energy level analysis. DLTS spectra (**a**) and Arrhenius plot (**b**) of the n-type silicon control sample by annealing the chemically modified silicon surface as shown in the inset. DLTS spectra (**c**) and Arrhenius plot (**d**) of the SAMM-doped Si (the SAMM structure is displayed in the inset). DLTS simulations on the spectra (rate window of 5 s$^{-1}$) of the control sample (**e**) and the SAMM-doped Si (**f**). Note that the DLTS signals in **e** are much smaller in amplitude than those in **f**

doped sample, blue curve), the kink at 75 K is absent (Fig. 4d inset), whereas the peak at 155 K and the broad bump both grow much bigger than the corresponding peaks in the control sample, probably due to the increase in defect concentration brought by extra amount of carbon and phosphorus. What is more, the shape of the broad bump is skewed in comparison with the control sample, clearly because the closely spaced peaks in the bump increase differently in amplitude. By comparing the three curves in Fig. 4d, we conclude that the SAMM doping process produces defects in phosphorus-doped silicon.

A better explanation for these phenomena needs quantitative identification of energy levels associated with the peaks. To find out the defect energy levels, DLTS measurements at different rate windows were carried out. It is known that DLTS signals peak when the charge emission rate from defects matches the experimental rate window given by the sampling time $t_1$ and $t_2$ (Supplementary Note 5). A higher rate window corresponds to a larger emission rate $e_n$, shifting DLTS peaks to higher temperatures (Fig. 5a, c), since the emission rate $e_n$ is correlated to temperature

$T$ and defect energy level $E_a$ by the following equation[17]:

$$e_n = (\sigma_n \langle v_n \rangle N_c / g) \exp\left(-\frac{E_a}{kT}\right) \quad (3)$$

where $\sigma_n$ is the capture cross-section, $\langle v_n \rangle$ is the mean thermal velocity of electron, $g$ is the degeneracy factor (chosen 2 here), $N_c$ is the effective density of states related to the semiconductor band structure, and $k$ is the Boltzmann constant.

Note that the factor $\langle v_n \rangle N_c$ is proportional to $T^2$. Hence, the logarithm term $\ln(e_n/T^2)$ is linearly correlated to $1/(kT)$, as shown in the Arrhenius plot of Fig. 5b, d. The slope of the lines gives the defect energy level $E_a$ and the intercept with $y$ axis provides information on the capture cross-section $\sigma_n$ (Supplementary Note 6 and Supplementary Table 2). In Fig. 5a (DLTS spectra of the control sample), two isolated peaks at low temperature range are detected. As the rate window increases, the peaks are right-shifted in the range from 60 to 80 K and from 130 to 160 K. The associated defect energy levels are determined to be 102 meV and

**Table 2 Comparison of the energy levels derived from DLTS spectra, simulations and energy levels of $C_i$–$P_s$ from ref. [32]**

|  | Bias pulse | Isolated peaks (meV) | | | Bump area (meV) | | | |
|---|---|---|---|---|---|---|---|---|
|  |  | Peak 0 | Peak 1 | Peak 2 | Peak 3 | Peak 4 | Peak 5 | Peak 6 |
| Control sample | −2 to 0 V | 102 | 254 | 319 | 378 | 390 | 467 | 480 |
| SAMM -doped sample | −2 to 0 V | -- | 252 | 319 | 380 | 390 (405) | 469 | 480 |
|  | −2 to −1V | -- | 260 | 319 | 380 | 390 | 467 | 480 |
|  | −0.2 to 0.2 V | -- | 260 | 319 | 380 | 390 | 467 | -- |
|  | 0 to 0.2V | -- | 260 | 319 | -- | 390 | 467 | -- |
| $C_i$–$P_s$ (ref. [32]) |  | -- | 260 | 320 | 380 | 390 | -- | 480 |

"--"means no peak is detected or has been reported in that position. The underscored energy levels are derived from simulations

254 meV (Fig. 5b), respectively. Considering that the only species really involved in the control sample are possibly carbon-related defects as mentioned above, the defect energy level at 102 meV can be best ascribed to carbon interstitials[31], configuration of which is shown in Supplementary Figure 8. The defect energy level at 254 meV continues to appear in the SAMM-doped sample (252 meV) with a higher amplitude. A defect energy level at 319 meV is extracted for the SAMM-doped sample from the isolated peak shifting from 155 to 190 K in Fig. 5c. At the region of temperature above 200 K, two main peaks with associated energy levels at 378 meV and 467 meV can be identified from the bump for the control sample (Fig. 5a, b). Similarly, two energy levels at 405 meV and 469 meV are identified for the SAMM-doped sample in Fig. 5c, d. However, the broad bumps in the DLTS spectra (Fig. 5a, c), clearly consisting of multiple closely spaced peaks may even contain more than those identified main peaks.

To identify the peaks in the broad bumps more accurately, DLTS simulations were conducted according to the basic principle as illustrated below, and the results were displayed in Fig. 5e, f. For DLTS, capacitance transient starts at the end of excitation pulse and then decays exponentially in its simplest form. The amplitude of a single peak detected at a given rate window can be expressed as Eq. (4).

$$\Delta C = \Delta C_0(\exp(-e_n t_1) - \exp(-e_n t_2)) \tag{4}$$

where $\Delta C_0$ is the initial capacitance transient (capacitance transient at the end of excitation pulse), and $t_1$ and $t_2$ defines the rate window. The emission rate $e_n$ is given by Eq. (3).

For multiple defect levels, the DLTS signal can be written as Eq. (5) (refer to Supplementary Equation (11)).

$$\Delta C = \sum \Delta C_{0i}(\exp(-e_{ni} t_1) - \exp(-e_{ni} t_2)) \tag{5}$$

in which $i$ represents the $i$th defect.

Table 2 summarizes the defect energy levels of the control and the SAMM-doped samples. The energy levels in bold are derived for the bias pulse from −2 to 0 V from both Arrhenius plots (number without underline) and simulation results (number with underline). The rest are for the other bias pulses, meaning that the DLTS are probing other regions, which will be discussed later. All the energy levels are in comparison with those of interstitial-carbon–substitutional-phosphorus ($C_i$–$P_s$) pairs from literature (the last row in Table 2). Five out of six energy levels for the SAMM-doped sample are consistent with the energy levels of $C_i$–$P_s$ pairs reported previously. Only peak 5 at the energy level near 467 meV or 469 meV is found independently in both control and SAMM-doped samples, suggesting that this defect energy level does exist despite not showing in ref. [32]. It is probably due to N-related defects[16] rather than $C_i$–$P_s$ multi-configurable defects, since the activation agent (DCC) and solvent (dimethylfoma-mide) in the SAMM grafting process contain nitrogen. Peak 4 in the SAMM-doped sample is determined by simulations to be 390

meV instead of 405 meV as shown in the Arrhenius plot in Fig. 5d. The peak at 405 meV indicated by the arrow in Fig. 5f is the result of overlap between peak 4 and peak 5. Note that the control sample is n-type silicon with phosphorus-doping concentration of $3 \times 10^{15}$ cm$^{-3}$ as purchased. Therefore, all the $C_i$–$P_s$ related energy levels shown up in the SAMM-doped sample also appear in the control sample (but with much smaller magnitude), because the carbon defects can bind with both the SAMM-introduced phosphorus dopants and the background phosphorus dopants in the n-type Si substrate (Fig. 4d). It is worth pointing out that the DLTS envelope by simulations does not match the experimental results perfectly. Some other peaks clearly exist, which may originate from surface states, nitrogen con-taminants[33], or atomic disorder[34]. The full deconvolution of the DLTS spectra can be found in Supplementary Figures 9–11 and Supplementary Table 3.

To show clearly that carbon defects bind with phosphorus dopants introduced by the SAMM doping process, we tuned the bias voltages from −2 to 0 V and injection pulses from −1 to 0.2 V, pushing the DLTS probing region from bulk to near the sur-face (Fig. 6)[35–37]. For comparison, SIMS profiling was also per-formed for phosphorus and carbon in both the SAMM-doped sample and the blank sample, as shown in Fig. 6a. The back-ground phosphorus doping ($3 \times 10^{15}$ cm$^{-3}$) is detected by CV technique (pink curve) but not by SIMS (blue curve and green triangles) due to the relatively high detection limit of SIMS. A combination of SIMS and CV measurements indicates that the SAMM-introduced phosphorus dopants has a concentration of around $2 \times 10^{18}$ cm$^{-3}$ near the surface and rapidly declines to the background doping concentration of $3 \times 10^{15}$ cm$^{-3}$ at a depth of 300 nm below the surface. The certificated carbon concentration in our n-type Si substrate is $<5 \times 10^{16}$ cm$^{-3}$. The carbon SIMS data reaches a floor at $2 \times 10^{16}$ cm$^{-3}$ in both the SAMM-doped sample and the blank sample, meaning that the actual back-ground carbon concentration in the substrate is at this level or even lower. The concentration of carbon impurities introduced by the SAMM doping process is ~$2 \times 10^{18}$ cm$^{-3}$ at the surface but slowly decays to $2 \times 10^{16}$ cm$^{-3}$ at a distance of about 300 nm from the surface. When the bias pulses to 0.2 from 0 V bias at 300 K, the depletion region edge sweeps approximately from 100 to 64 nm below the surface, in which carbon and phosphorus impu-rities mainly come from the SAMM, as shown in Fig. 6b. Note that the depletion region moves slightly deeper into the substrate at low temperature (Supplementary Figures 12 and 13 and Sup-plementary Table 4). Fig. 6c depicts the corresponding DLTS data within the above sweep range of the depletion region. All the peaks shown here are included in Table 2. DLTS is repeated at other bias pulses. The probing range and corresponding DLTS data are shown in Fig. 6d–i (also see the defects energy levels in Table 2). A detailed analysis on the peak positions and amplitudes will not be conclusive due to the well-known metastability of carbon-related complex defects[32]. But overall the experimental

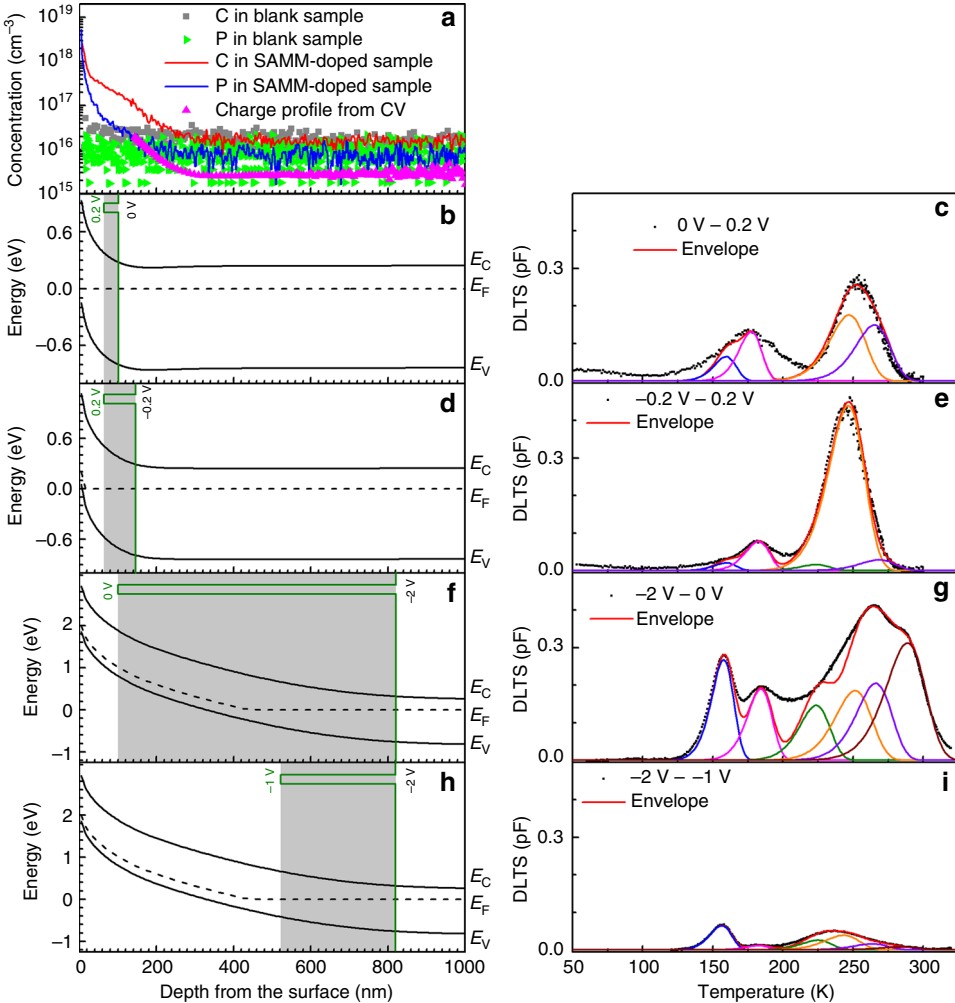

**Fig. 6** DLTS probing region analysis. **a** Phosphorus and carbon depth profiles by SIMS compared with ionized charge profile derived from CV. Silvaco simulation on band structure at 300 K with bias voltage of 0 V (**b**), −0.2 V (**d**), −2 V (**f**) and −2 V (**h**). Probing regions are shaded in gray with different pulses from 0 to 0.2 V (**b**), −0.2 to 0.2 V (**d**), −2 to 0 V (**f**) and from −2 to −1 V (**h**). DLTS simulations on the spectra of the SAMM-doped silicon with pulses from 0 to 0.2 V (**c**), −0.2 to 0.2 V (**e**), −2 to 0 V (**g**) and −2 to −1 V (**i**). The rate windows of DLTS spectra are 200 s$^{-1}$. Note that **c**, **e**, **g** and **i** have the same y axis scale for better comparison. A close-up figure for **i** to show the fitting envelope can be found in Supplementary Figure 11

observations are consistent with the fact that a larger quantity of carbon and phosphorus impurities in a probing region will lead to stronger DLTS signals. For example, though the probing region width in Fig. 6h is larger than that in Fig. 6d, the corresponding DLTS signals in Fig. 6i are much weaker because the DLTS is probing a region deep in the bulk (Fig. 6h) where the phosphorus and carbon concentration are much lower.

## Discussion

It is known that ultra-shallow junctions as the source and drain of modern complementary metal–oxide–semiconductor (CMOS) transistors help suppress the short channel effect[8]. The SAMM doping technique has the unique advantage of forming ultra-shallow junctions[8]. However, the ultra-scaled thickness of the junctions will inevitably increase series resistance in the source and drain, resulting in inferior performances for CMOS transistors. A possible solution is to increase the dopant concentration by increasing the molar ratio of dopant elements in the carrier molecule as demonstrated previously[38]. However, according to the carbon defect formation mechanism, a higher phosphorus concentration may lead to a larger portion of inactive phosphorus, offsetting the effect of higher dopant molar ratio on reducing the series resistance. Logically, new processes should be

developed to remove carbon in dopant carrying molecules prior to thermal annealing so that the $C_i$–$P_s$ defects can be minimized to achieve a high ionization rate for phosphorus dopants.

In conclusion, we have successfully doped silicon with phosphorus by SAMM doping technique via a two-step molecular monolayer grafting process. Phosphorus is incorporated into silicon with an areal dose of $1.34 \times 10^{13}$ cm$^{-2}$. However, only 80% ($1.07 \times 10^{13}$ cm$^{-2}$) of phosphorus is electrically active and the rest 20% is deactivated. Carbon diffuses into silicon together with phosphorus but with a much deeper depth. This carbon can bond with group V element forming complex defects. Corresponding deep energy levels are detected by DLTS for the first time in SAMM doping technique. With the assistance of DLTS simulation, multi-configurational defects $C_i$–$P_s$ are confirmed, indicating that phosphorus dopants are partially deactivated by interstitial carbon. Therefore, for SAMM doping technique, carbon in dopant-carrying molecules is recommended to be removed or controlled at low concentration before thermal annealing.

## Methods

**Materials**. FZ single-side polished silicon wafers, (100)-oriented (⟨100⟩ ± 0.05°), 500 ± 25 µm thick, >10 kΩ cm in resistivity, and CZ single-side polished silicon wafers (100)-oriented (⟨100⟩ ± 0.05°), n-type (phosphorus), 500 ± 10 µm thick, 1–3

$\Omega$ cm in resistivity, were purchased from Suzhou Resemi Semiconductor Co. Ltd., China. All chemicals, unless noted otherwise, were of analytical grade and used as received. Isopropanol, acetone, and ethanol for surface cleaning were of CMOS grade. 5-Hexenyl acetate (98%) was purchased from TCI, Shanghai. Mono-N-dodecyl phosphate (97%) was from Alfa Aesar. Dicyclohexylcarbodiimide (DCC, 99%), lithium aluminum hydride ($LiAlH_4$ powder, reagent grade, 95%), and hydrofluoric acid (HF, 48%, CMOS grade) were from Sigma Aldrich.

**Wafer cleaning**. Si wafers were cleaved into 1.5 cm by 1.5 cm pieces and cleaned with acetone and ethanol of CMOS grade in a sonication bath for 5 min, respectively. After rinsed with deionized (DI) water, the Si samples were immersed in "piranha solution" (98% $H_2SO_4$:30% $H_2O_2$, 3:1 (v/v)) for 30 min at 90 °C, followed by rinsing with DI water again. The wafers were then etched in 2.5% HF solution for 90 s to remove the oxide layer and render a hydrogen-terminated surface. The hydrogen-terminated samples were quickly rinsed in DI water, blown dry with nitrogen, and immediately proceeded to further modification.

**Thermal hydrosilylation and surface functionalization**. First, 5-hexenyl acetate was grafted onto Si by hydrosilylation reaction. The freshly etched Si (100) samples were immediately transferred to a deoxygenated sample of neat 5-hexenyl acetate in a dry Schlenk tube under Ar atmosphere. The reaction was then conducted at 95 °C in Ar atmosphere for 16 – 19 h. The resulting samples were copiously rinsed with ethanol, dichloromethane, and acetone, respectively, and then blown dried by a stream of $N_2$.

Subsequently, the acetate-terminated surface was immersed in dry tetrahydrofuran (THF) with 5% (w/v) $LiAlH_4$ and refluxed at 70 °C for 2 h. After rinsing with DI water and ethanol, the hydroxyl-terminated samples were immersed into 0.5 M hydrochloric acid for 20 min to remove any Al residues.

In the presence of bifunctional crosslinker DCC (40 mM), the hydroxyl-terminated samples were reacted with mono-dodecyl phosphate (5 mM) in dimethylfomamide (DMF) at room temperature for 48 h, affording phosphorus-containing functionalization. The samples were washed carefully with ethanol, dichloromethane, and acetone to remove any remaining coupling reagents and dried under $N_2$ stream for further treatments.

**Silicon dioxide deposition and thermal annealing**. $SiO_2$ capping layers on silicon were produced by SOG method with IC1-200 polysiloxane-based coating material (Futurrex Inc. USA). Briefly, silicon wafer was spin-coated with the IC1-200 at 3000 rpm for 40 s, followed by 100 °C bake on a hot plate for 60 s, 200 °C for 60 s and 400 °C bake in Ar for 30 min. After the formation of the capping layers, the functionalized silicon samples were thermally annealed at 1050 °C for 120 s, with a ramp temperature of 100 °C min$^{-1}$, starting from 800 °C, in Ar environment in a tube furnace (Thermo scientific Lindberg/Blue, USA). After annealing, the doped Si samples were immersed in BOE (buffer oxide etchant) solution (HF:$NH_4F$ = 6:1, CMOS grade, J.T. Baker Co. USA) to remove $SiO_2$ layer.

**Surface characterization**. XPS was carried on a Kratos AXIS UltraDLD spectrometer with a monochromated Al K$\alpha$ source (1486.6 eV), a hybrid magnification mode analyzer and a multichannel detector at a takeoff angle of 90° from the plane of the sample surface. Analysis chamber pressure is <$5 \times 10^{-9}$ Torr. All energies are reported as binding energies in eV and referenced to the C 1s signal (corrected to 285.0 eV) for aliphatic carbon on the analyzed sample surface. Survey scans were carried out selecting 250 ms dwell time and analyzer pass energy of 160 eV. High-resolution scans were run with 0.1 eV step size, dwell time of 100 ms and the analyzer pass energy set to 40 eV. After background subtraction using the Shirley routine, XPS spectra were fitted with a convolution of Lorentzian and Gaussian profiles by using software Casa XPS. Secondary-ion mass spectrometry (SIMS) was conducted to obtain dopant profile at the top 500 nm of substrate by Evans Analytical Group, NJ, USA.

**Van der Pauw and Hall measurements**. The metal contacts on silicon for electrical measurements were realized by evaporating 200-nm aluminum or aluminum/gold films in a thermal evaporation system (Angstrom Engineering, Canada). Van der Pauw measurements were performed on square-shaped samples on which the metal contacts are exactly located at the four corners. The custom-made probe station shrouded in a completely dark metal box is equipped with four solid tungsten probe tips (the tip size < 1 μm). Keithley 2400 source meter units and a custom-written Labview script were employed to generate and collect current/voltage data. Hall measurements were performed on the same square-shaped samples, which were pre-mounted onto a dc resistivity sample holder via wire bonding, in a Physical Property Measurement System (PPMS, Quantum Design, USA).

**DLTS**. Schottky diodes for DLTS measurement are fabricated by depositing a circle Au electrode of 1 mm in diameter and 150 nm thick on the top of silicon and 150 nm thick Al on the backside via thermal evaporation (Angstrom Engineering, Canada). The circle Au electrode is deposited with assistance of lithography technique. The diodes are then mounted onto sample holder TO5. A conventional

DLTS with boxcar mode are applied to get better resolution. Data were collected using Laplace DLTS software and plotted in usual DLTS plots.

**SIMS**. The carbon and phosphorus SIMS profiling was conducted at EAG laboratories in USA under high vacuum condition of about $3 \times 10^{-11}$ Torr. Focused Cs$^+$ primary ion beam was applied for sputtering, which facilitate high yields of secondary ions of phosphorus and carbon. Before sputtering, the samples were cleaned with oxygen plasma to remove possible carbon contamination from air.

**Data availability**. The data that support the findings of this study are available from the authors on reasonable request, see author contributions for specific data sets.

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

## Acknowledgements

The work is supported by the national "1000 Young Scholars" program of the Chinese central government, the National Science Foundation of China (grant number 21503135), the SJTU-UM Collaborative Research Program and the "Innovative Research Plan" of the Shanghai Bureau of Education. XPS and Hall Effect measurements are performed at Instrumental Analysis Center (IAC) and some microfabrication processes are carried out at the Center for Advanced Electronic and Material Devices (AEMD), Shanghai Jiao Tong University (SJTU). The authors appreciate Dr. Limin Sun and Ligang Zhou at IAC of SJTU for valuable discussions about XPS and Hall Effect measurements.

## Author contributions

Y.D. conceived the idea and directed the research. X.G., B.G. and Y.D. wrote the manuscript. X.G. and B.G. prepared the samples and carried out the electrical characterizations. A.M. performed DLTS measurements. X.G. and A.M. analyzed the DLTS data. K.C. simulated the energy band bending and redistribution of electrons ionized from phosphorus dopants. All authors reviewed the manuscript.
