## [Peer Review File · Nature Communications]

Reviewers' Comments:

Reviewer #1 (Remarks to the Author)

Doping by self-assembled molecular monolayers is a promising and controversial method to replace conventional doping of silicon by a more regular and low cost doping. Several laboratories are considering this method but its reliability has not definitively been proved.

This work makes a major step to identify significant drawbacks of the method, namely formation of C related traps incorporating P dopants. Therefore, it paves the way to future improvements to make it free from unwanted effects. The research method is solid and full information of how the conclusion have been achieved is provided. I recommend publication of this paper on Nature Communication as is.

Minor revisions: I suggest to replace or integrate Ref. 26 with a more exhaustive version, namely:

Altermatt, P. P., A. Schenk, and G. Heiser. "A simulation model for the density of states and for incomplete ionization in crystalline silicon. I. Establishing the model in Si: P." *Journal of Applied Physics* 100.11 (2006): 113714

Reviewer #2 (Remarks to the Author)

Authors have made a very interesting study of how phosphorus (P) dopes silicon using the self-assembled molecular monolayer technique. It is a topic worth for nano communications, however, the manuscript should go through major revision.

I have several concerns regarding their experimental procedure, specially the electrical characterization.

To start with, in the abstract they mention a carbon-related defect concentration of $2.7 \times 10^{12} \text{ cm}^{-2}$ but it is nowhere measured. It is in page 7 where they speculate it is related to the electrically unactivated phosphorus dopants. This is pure speculation and they can not conclude that is the carbon concentration.

Also, through the article, they use the term "monolayer-doped silicon"... this cause me problems when reading it, specially when I saw Fig. 2, S3 and S4, where clearly P concentration is not a monolayer... after some time I realized that what they meant with this "monolayer-doped silicon": they meant the Si sample that was doped via self-assembled molecular monolayer technique. Authors should be careful with this, because there is a monolayer doping technique where definitely it is ONE monolayer that is grown inside a sample, like for sample Sb-delta doping in Silicon. I would advice that at the beginning of their articles, the authors should mention something like that from now on the Si sample that was doped using the self-assembled molecular monolayer technique will be labeled SAMM-doped-Si, o SAMM-Si, or something like that.

Regarding their van der Pauw measurements, at the end of their article they mention they used 200 nm Al or Al/Au contacts. These are very bad candidates to make ohmic contacts on n-type Si, because Al diffuses a lot into Si and it is used for making ohmic contacts but on p-type Si. They also dont mention if there was any thermal treatment which is necessary usually and they do not show any IV curve showing good ohmic behaviour. They should repeat this part using Au contacts and thermally treating them at 360oC for at least 5 min in N2 environment, as this form the eutectic and good ohmic contacts on Si. They should also show the IV curves between each pair of contacts, showing linear behaviour, otherwise the van der Pauw method can not be used. Also, where the van der Pauw measurements done with room light or in darkness? They should clarify it.

In Page 3 they say "freshly prepared hydrided-terminated silicon"... they should give more details and/or some reference.

Regarding their Scheme 1, it really confused me the inset with the molecules on top of the figure of Surface 1, it looks like it is related to surface 1... I advice to put it left of Surface 1 mentioning it is the schematic figure of the molecules.

In page 4, they say "buffered oxide etchant"... what was it? Hydrofluoric acid? they should clarify this.

As they mention at the end of page 5 and beginning of page 6 the phosphorus dopant distribution is highly non-uniform. Thus, they need to use a several layers model to obtain the total measured Hall resistance, the way they proceed is wrong, because they proceed considering an homogeneous concentration, which is not the case (Fig. 2, S3 and S4). Also, why on these figures there are two point values close to zero which much more smaller values? they should explaining this.

They should put a reference for their Eq. 1, and as discussed on the former paragraph, Eq. 1 can not be used, as it is for the case of an homogeneous doping.

Regarding their Fig.2 and S4, I think they should change them just for the insets, which give more information as the y-axis is in log scale, which is better. Also, when doing this, pay attention to inset of S4, as the y-axis is in engineer plot, not scientific units.

Also, their Fig. 2 b, in the x-axis it says magnetic flux density with Tesla units!!!! What did the authors do? measured the magnetic field or the magnetic flux? According to their Eq. 1, it seems they measure magnetic field, which is the one they need, then they have to correct Fig. 2 b, put magnetic field and then units can be Tesla.... but if they measured magnetic flux, then they have to explain how it is related to magnetic field and use weber or maxwell units in Fig. 2 b.

Also in Page 7, they say "we chose an n-type Si (100) substrate... to confine the Schottky depletion region near the surface". This confused me for a while, as if they somehow modified the original sample putting some kind of this layer on it.... after a while I realized they prepared a new sample... this should be clarified with some sentence like "in order to have a Schottky contact confined close to the surface, we prepare a new sample....".... remember to use some name like SAMM-S4

I have concerns about their use of Au as a Schottky diode in Silicon. This is a terrible experimental choice, as Au diffuses very quickly into Si and forms an electron level with energy around 280 meV and a hole level with 480 meV, which they should clearly see and/or affect their DLTS measurements. I consider their DLTS results can not be presented like this, and they should repeat the Schottky diode formation using a Titanium Schottky diode, as Ti is one of the best choices in this case. Also regarding the Schottky diode, they mention they used Al for the back ohmic contact. As mentioned before, this is a bad choice, and they should use Au and anneal it. See the fifth paragraph in this discussion.

When they manage to produce this kind of Schottky diodes, they should report also capacitance voltage measurements and do an analysis... they should find a charge carrier density profile similar to their Fig. 2, S3 and S4.

In page 8 they mention "with a high degree of confidence we would attribute most of the observed defects to complexes where carbon is the main ingredient"... this sound to me with too much confidence and very little evidence.... they should be able to see the Carbon profile in their SIMS measurements... why dont they see it?

In page 9, they should clarify how the rate windows related to the charge emission rate to obtain the peak on the DLTS signal.

In the supplementary material, the DLTS simulation section, it is the only place where they mention the reverse and pulse bias condition, being $-2\text{ V} / -0.5\text{ V}$ and $-2\text{ V} / -1\text{ V}$ I have concerns with this values. The capacitance voltage measurements that they have not done and report, could tell us what is the depletion region at this voltage values.... they should report this to make sure that they are studying the region they claim to study, i.e., the first 200 nm from the surface. Also, what was the time pulse?

Also, in their simulations Fig. 5, it looks to me they need more peaks to fit well the curves.

In the DLTS section page 15, they mention the data was collected using Laplace DLTS software.... so are their DLTS curves, usual DLTS plots or Laplace DLTS plots?

In summary, I think the topic of research is interesting and suitable for nanocommunications. However the authors need to correct the electrical characterization part. Briefly:

1. Repeat van der Pauw measurements using the ohmic contacts mentioned here.
2. Show IV curves that prove they made suitable ohmic contacts.
3. Redo the Schottky devices, using Ti for Schottky and annealed Au for the back ohmic contact. Remember to do FIRST the back contact and SECOND the Schottky contact. Clarify this clearly when you do it.
4. Make IV and CV curves for the Schottky diode, at several temperatures.
5. Specially analyse the CV curves to clearly determine the voltage parameters for the DLTS measurements such that they really study the first 200 nm layer from the surface where they claim the P-doped the sample.
6. Redo SIMS and try to see carbon profiling.
7. Correct all the other minor details that have been mentioned.

Good luck!

Response Letter to Reviewers' comments on

Deep level transient spectroscopic investigation of phosphorus-doped silicon by self-assembled molecular monolayers

Xuejiao Gao^{1†}, Bin Guan^{1†}, Abdelmadjid Mesli², Kaixiang Chen¹ and Yaping Dan^{1*}

¹University of Michigan – Shanghai Jiao Tong University Joint Institute, Shanghai Jiao Tong University, Shanghai, 200240, China

²Institut Matériaux Microélectronique Nanosciences de Provence, UMR 6242 CNRS, Université Aix-Marseille, 13397 Marseille Cedex 20, France.

[†] These authors contributed equally to this work

* To whom correspondence should be addressed. Email: yaping.dan@sjtu.edu.cn

Reviewers' comments:

Reviewer #1 (Remarks to the Author):

Doping by self-assembled molecular monolayers is a promising and controversial method to replace conventional doping of silicon by a more regular and low cost doping. Several laboratories are considering this method but its reliability has not definitively been proved.

This work makes a major step to identify significant drawbacks of the method, namely formation of C related traps incorporating P dopants. Therefore, it paves the way to future improvements to make it free from unwanted effects. The research method is solid and full information of how the conclusion have been achieved is provided. I recommend publication of this paper on Nature Communication as is.

Response: We really appreciate the reviewer for these very positive comments.

Minor revisions: I suggest to replace or integrate Ref. 26 with a more exhaustive version, namely:

Altermatt, P. P., A. Schenk, and G. Heiser. "A simulation model for the density of states and for incomplete ionization in crystalline silicon. I. Establishing the model in Si: P." *Journal of Applied Physics* 100.11 (2006): 113714

Response: We thank the reviewer for this suggestion. The suggested reference has been added as Ref. 26.

Reviewer #2 (Remarks to the Author):

Authors have made a very interesting study of how phosphorus (P) dopes silicon using the self-assembled molecular monolayer technique. It is a topic worth for nano communications, however, the manuscript should go through major revision.

Response: We sincerely thank the reviewer for recommending the manuscript to publish in Nature Communications. We will respond to the raised questions one by one after we divide the reviewer's comments sequentially into the following 21 sections.

I have several concerns regarding their experimental procedure, specially the electrical characterization.

(1). To start with, in the abstract they mention a carbon-related defect concentration of $2.7 \times 10^{12} \text{ cm}^{-2}$ but it is nowhere measured. It is in page 7 where they speculate it is related to the electrically unactivated phosphorus dopants. This is pure speculation and they cannot conclude that is the carbon concentration.

Response: Considering that there is some uncertainty in Hall measurements due to non-uniform distribution of dopants as the reviewer commented in (7) below, we agree with the reviewer and have removed this number from the abstract.

(2). Also, through the article, they use the term "monolayer-doped silicon"... this cause me problems when reading it, specially when I saw Fig. 2, S3 and S4, where clearly P concentration is not a monolayer... after some time I realized that what they meant with this "monolayer-doped silicon": they meant the Si sample that was doped via self-assembled molecular monolayer technique. Authors should be careful with this, because there is a monolayer doping technique where definitely it is ONE monolayer that is grown inside a sample, like for sample Sb-delta doping in Silicon. I would advise that at the beginning of their articles, the authors should mention something like that from now on the Si sample that was doped using the self-assembled molecular monolayer technique will be labeled SAMM-doped-Si, o SAMM-Si, or something like that.

Response: We thank the reviewer for reminding us of this potential confusion. Accordingly, we have replaced the term "monolayer-doped silicon" with "self-assembled molecular monolayer (SAMM) doped Si" in the revised manuscript.

(3). Regarding their van der Pauw measurements, at the end of their article they mention they used 200 nm

Al or Al/Au contacts. These are very bad candidates to make ohmic contacts on n-type Si, because Al diffuses a lot into Si and it is used for making ohmic contacts but on p-type Si. They also don't mention if there was any thermal treatment which is necessary usually and they do not show any IV curve showing good Ohmic behaviour. They should repeat this part using Au contacts and thermally treating them at 360°C for at least 5 min in N₂ environment, as this form the eutectic and good Ohmic contacts on Si. They should also show the IV curves between each pair of contacts, showing linear behavior, otherwise the van der Pauw method cannot be used. Also, where the van der Pauw measurements done with room light or in darkness? They should clarify it.

Response: We cannot agree with the reviewer on this point. According to the table listed on Page 190 in the SZE textbook (Physics of Semiconductor Devices, 3rd ed., S.M. Sze and Kwok K. Ng, John Wiley & Sons, Inc, 2007), Al makes good Ohmic contact on n-type silicon and might form bad Schottky contact on p-type Si, depending on the surface quality. Instead, Au will form good Ohmic contact on p-type Si and good Schottky contact on n-type Si (see the same table in the SZE textbook).

Our as-purchased wafer in DLTS analysis is n-type Si (phosphorus doped). A high concentration of phosphorus dopants (along with carbon) is introduced by the SAMM into the wafer from the polished Si surface. Au microelectrodes (without the adhesion layer such as Cr or Ti) are deposited on this polished surface by thermal evaporation, after the surface is treated with HF. A good Schottky contact is formed between Au and n-Si without thermal annealing (see the I-V curve in Fig.3a in the manuscript).

For the backside contact, we first scratched the surface extensively using a diamond scribe. The scratched area is treated with HF (using Q-tip) before Al evaporation. No thermal annealing is performed. The scratching will create a lot of defects in silicon which will “destroy” any possible small barrier between Si and Al. After this treatment, a good Ohmic contact is formed between Al and n-type Si, which is self-illustrative in the exponential I-V curve at positive bias in Fig.3a. To further prove that Al does form good Ohmic contact with n-type Si, we repeated the experiments on the backside of the same n-type Si wafer. Two Al pads were formed on the scratched areas (after HF treatment) by thermal evaporation using an Al foil shadow mask. The optical image of the sample is shown in the inset of the following Fig.A. Without thermal annealing, the I-V curve is perfectly linear as the bias voltage sweeps from -2V to 2V.

For the Al contacts in van der Pauw measurements shown in Table 1 in the manuscript, the process is a little different. No scratching is conducted. After thermal evaporation of Al, thermal annealing (370°C for 1min in Ar) is often required to form Ohmic contact. I-V curves between each pair of electrodes on the samples under investigation are shown in Fig.B1, B2 and B3. The curves for most contacts are perfectly linear, except some contacts on the as-received intrinsic Si wafer showing slight nonlinearity. The

linearity of the I-V curves indicates that Ohmic contacts are formed. The measurements were performed in a homemade probe station shrouded in a completely dark metal box.

Accordingly, we add a sentence on Page 8 to emphasize that no thermal annealing is performed on the sample for DLTS measurements. We also clearly state that van der Pauw measurements were performed in darkness on Page 4. Lastly, we added Fig. A and B1, B2, B3 and the related discussions to the supporting information.

Figure A. I-V curve proving good Ohmic contact of Al electrode on n-type Si wafer. Inset is the optical image of the testing sample, on the backside of which two areas are firstly scratched and then deposited with Al electrodes after HF treatment. Note that the yellow-ish color of the Al electrodes is due to the vacuum tape (yellow-ish) used to ensure the Al foil shadow mask in contact with the substrate.

Figure B1. I-V curves of unmodified sample (as-received intrinsic-Si wafer after cleaning)

Figure B2. I-V curves of control sample (monolayer without phosphorus)

Figure B3. I-V curves of SAMM-doped sample

(4). In Page 3 they say "freshly prepared hydrided-terminated silicon"... they should give more details and/or some reference.

Response: According to the wafer cleaning description in experimental section, the wafer was treated lastly with 2.5 % HF for 90 s. In this process, the oxide layer formed on the silicon wafer is etched by HF, rendering a hydrogen-terminated surface. To make it clear, the sentence in Line 1 Page 14 for describing the wafer cleaning process has been modified to "The wafers were then etched in 2.5% HF solution for 90 seconds to remove the oxide layer and render a hydrogen-terminated surface. The hydrogen-terminated samples were quickly rinsed in DI water, blown dry with nitrogen and immediately proceed to further modification"

(5). Regarding their Scheme 1, it really confused me the inset with the molecules on top of the figure of Surface 1, it looks like it is related to surface 1... I advice to put it left of Surface 1 mentioning it is the schematic figure of the molecules.

Response: We thank the reviewer for bringing this to our attention. We have moved molecule 2 to the top of surface 3 as shown below. The figure in the revised manuscript has been updated.

Scheme 1. Stepwise surface modification on Si (100) surfaces. Molecule 1 is chemically grafted onto surface 1 under thermal treatment at 95 °C for 16 h, forming a molecular monolayer. Molecule 2 reacts with the hydroxyl group on surface 3, leading to a phosphorus-functionalized surface 4.

(6). In page 4, they say "buffered oxide etchant"... what was it? Hydrofluoric acid? They should clarify this.

Response: Buffered oxide etchant is the standard name for the mixture of HF and NH₄F widely used in nano/microfabrication facilities. To avoid any possible confusion, we have modified the sentence as “The SiO₂ layer was later removed by BOE (buffered oxide etchant, HF:NH₄F = 6:1) before electrical characterizations.”

In the experimental section, changes are also made: “After annealing, the doped Si samples were immersed in BOE (buffered oxide etchant) solution (HF:NH₄F=6:1, CMOS grade, J.B. Baker Co. USA) remove SiO₂ layer.”

(7). As they mention at the end of page 5 and beginning of page 6 the phosphorus dopant distribution is highly non-uniform. Thus, they need to use a several layers model to obtain the total measured Hall resistance, the way they proceed is wrong, because they proceed considering a homogeneous concentration, which is not the case (Fig. 2, S3 and S4).

Response: We thank the reviewer for bringing this to our attention. It is true that the multilayer model will be more accurate than the uniform model. The Hall coefficient of the multiple layer model is given on Page 350 in “The Hall effect and its applications [M]. Springer Science & Business Media, 2013” as following:

$$R = \frac{te \int_0^t n(x)\mu^2(x)dx}{\left[e \int_0^t n(x)\mu(x)dx \right]^2}$$

where t is the thickness of the sheet, e the unit electron charge and $\mu(x)$ the charge carrier mobility.

One of the empirical mobility models in literature is given in “Ho J C, Yerushalmi R, Jacobson Z A, et al. Controlled nanoscale doping of semiconductors via molecular monolayers. Nature materials, 2008, 7(1): 62-67” as following:

$$\mu = \mu_{min} + \frac{\mu_{max} - \mu_{min}}{1 + \left(\frac{N}{N_r}\right)^\alpha}$$

The surface electron concentration is defined as:

$$n(t) = \int_0^t n(x)dx$$

Using the uniform model for Hall Effect measurements, the measured electron surface concentration at room temperature is $8.92 \times 10^{12}/cm^2$ as shown in the manuscript. The concentration calculated by the multilayer model is $9.7 \times 10^{12}/cm^2$. This indicates that the uniform model used in the original manuscript underestimates the electron concentration by only $\sim 8\%$. This relatively small error is likely because the dopants in our case, although non-uniform, are mostly located within a very thin layer ($<200nm$). This error will not change the conclusion of this manuscript.

However, let’s do not forget that our dopants are mostly located near the surface and co-doped with a lot of carbon. It is very difficult to ensure the empirical mobility model (for bulk doping) or the measured mobility values in literature can accurately predict (say less than 10% error) the actual mobility in our sample. The use of the mobility values that are of high uncertainty will only render even bigger uncertainty in the calculated electron concentration.

As a result, we decide to keep eq.(1) as it is, but are happy to put a note in the manuscript that the model may lead to a few percent errors in actual electron concentration. The above discussion has been added into supplementary information.

(8). Also, why on these figures there are two point values close to zero which much smaller values? They should explaining this.

Response: It is known that first few points near interface in SIMS data are usually inaccurate due to technical issue (see explanation below). This phenomenon is widely observed in literature, such as Fig 2

in the paper of Shimizu et al (Nanoscale 2014, 6, (2), 706-710) and Fig 3(a) in the paper of Ho et al (Nature materials 2008, 7, (1), 62-67).

In SIMS analysis, the element concentration is calculated according to the equation below:

$$C_a = RSF \times \frac{I_a}{I_s}$$

where C_a is the concentration of element a, RSF is relative sensitivity factor, I_a is the intensity of ionized element a, I_s is the intensity of ionized substrate. For the first two or three steps of sputtering, the material is not pure silicon (for example, native oxide), but engineers usually use RSF of silicon to do calculation.

In our case, since the dopants diffuse far into the substrates (hundreds of nanometers), the error for the first few data points from this technical issue is negligible.

To avoid confusion, we inserted the above discussion in the supporting information.

(9). They should put a reference for their Eq. 1, and as discussed on the former paragraph, Eq. 1 cannot be used, as it is for the case of a homogeneous doping.

Response: As we responded in (7), the big uncertainty in mobility will result in an even bigger error in electron concentration found from the multilayer Hall Effect model, although the multilayer model will be more accurate for non-uniform doping. In fact, the uniform doping model we use will result in only a few percent errors in the actual electron concentration, which will NOT affect the conclusion we reached in the manuscript. As a result, it is reasonable to keep the eq.(1) as it is. But we are happy to acknowledge in the manuscript that the uniform model may lead to a few percent errors in electron concentration.

(10). Regarding their Fig.2 and S4, I think they should change them just for the insets, which give more information as the y-axis is in log scale, which is better. Also, when doing this, pay attention to inset of S4, as the y-axis is in engineer plot, not scientific units.

Response: We really appreciate the reviewer for giving us the suggestion to present the figures better. We have made the changes accordingly, as shown below and also in the manuscript and SI.

Fig.2

Fig. S4

(11). Also, their Fig. 2 b, in the x-axis it says magnetic flux density with Tesla units!!!! What did the authors do? Measured the magnetic field or the magnetic flux? According to their Eq. 1, it seems they measure magnetic field, which is the one they need, then they have to correct Fig. 2 b, put magnetic field and then units can be Tesla.... but if they measured magnetic flux, then they have to explain how it is related to magnetic field and use weber or maxwell units in Fig. 2 b.

Response: We truly thank the reviewer for bringing this to our attention. This is our mistake which has been corrected in the revised manuscript accordingly.

(12). Also in Page 7, they say "we chose an n-type Si (100) substrate... to confine the Schottky depletion region near the surface". This confused me for a while, as if they somehow modified the original sample putting some kind of this layer on it.... after a while I realized they prepared a new sample... this should be

clarified with some sentence like "in order to have a Schottky contact confined close to the surface, we prepare a new sample...." remember to use some name like SAMM-S4

Response: Thank the reviewer for reminding us about this. We were not aware that readers might get confused. We agree with the reviewer. The sentence has been modified as "To detect possible defects in this region, we prepared a new sample on n-type Si (100) substrate (phosphorus-doped) with resistivity of 1~3 $\Omega\cdot\text{cm}$ (doping concentration $1\sim 5 \times 10^{15}/\text{cm}^3$) to confine the Schottky depletion region near the surface. The same SAMM-doping process as described previously (on SAMM-doped surface 4) was conducted on the n-type substrate."

(13). I have concerns about their use of Au as a Schottky diode in Silicon. This is a terrible experimental choice, as Au diffuses very quickly into Si and forms an electron level with energy around 280 meV and a hole level with 480 meV, which they should clearly see and/or affect their DLTS measurements.

Response: We cannot agree with the reviewer. Au is in fact a good choice for forming Schottky diode on n-type Si as we discussed in (3) previously. The DLTS samples did not went through any thermal annealing. Schottky diode between Au and n-Si and Ohmic contact between Al and n-Si are formed at room temperature. The diffusion coefficient of Au in silicon is given by: $D(\text{cm}^2/\text{s}) = 0.021 \exp(-1.7(\text{eV})/kT)$ [Coffa et al. JAP 64, 6291-5 (1988)]. To have an appreciable fraction of Au diffusing into the sample, the temperature must be very high, which is not the case in the present work. As a result, the reviewer's concern on Au diffusion and contamination is not justified.

In fact, the reviewer's comments on Au defect energy level are inaccurate, although it is not related to the manuscript. Au induces two levels indeed: an acceptor and a donor state. The acceptor state acts as a generation – recombination center. It is located near mid band gap (0.55 eV) and thus acts as both electron and hole trap. The donor state is exclusively a hole trap (not electron trap) and is located at 0.28 eV above the valence band. The acceptor state can in some circumstances be seen in p-type silicon, that is, as a hole trap. We may provide references for that although this is out of subject here as we are dealing exclusively with n-type Si.

(14). I consider their DLTS results cannot be presented like this, and they should repeat the Schottky diode formation using a Titanium Schottky diode, as Ti is one of the best choices in this case. Also regarding the Schottky diode, they mention they used Al for the back ohmic contact. As mentioned before, this is a bad choice, and they should use Au and anneal it. See the fifth paragraph in this discussion.

Response: Our previous discussions in (3) and (13) have made it clear. No further response to this comment.

(15). When they manage to produce this kind of Schottky diodes, they should report also capacitance voltage measurements and do an analysis... they should find a charge carrier density profile similar to their Fig. 2, S3 and S4.

Response: we measured CV at room temperature and derived the carrier concentration profile, as shown in Fig.C below. However, it is difficult to measure the profile in 3 Debye lengths near the surface due to the limitation of depletion region under 0 V on the analyzed sample. More discussion to this point can be found in (18). Overall, CV offers less information than DLTS and low-temperature Hall measurement in this research.

Figure C. (a) C-V curves and (b) ionized charge profiles for P-doped sample and control sample measured at room temperature. Both of samples were measured with pulse at 1MHz.

The following paragraph has been added in supplementary information.

Capacitance-voltage curves were measured on the control sample and phosphorus-doped sample with pulses at 1MHz at room temperature, as shown in Fig. C (a). Ionized charge profiles are derived from the C-V curves, shown in Fig. C(b). The ionized charge profile of the P-doped sample is higher than the control sample, demonstrating successful incorporation of P by SAMM process. With reversed-bias varying from 0V to -2V, detection region of P-doped sample ranges from 307 nm to 663 nm, while control sample ranges from 469 nm to 999 nm. However, it is difficult to measure the charge carrier profile in 3 Debye

lengths near the surface using this technique, as conventional DLTS on Schottky device only investigate the area beyond the depletion region under 0V (which is roughly the top 300 nm in our SAMM-doped sample).

(16). In page 8 they mention "with a high degree of confidence we would attribute most of the observed defects to complexes where carbon is the main ingredient".... this sound to me with too much confidence and very little evidence.... they should be able to see the Carbon profile in their SIMS measurements... why don't they see it?

Response: Indeed, the carbon profile should be recorded. To get the carbon profile, we send two samples to Evans Analytical Group (EAG) to redo SIMS. The first one is the SAMM-doped sample and the second one is a blank sample (as-received Si wafer) as a control. The data are shown in Fig.D.

Figure D. Black: Carbon profile in the blank sample. Carbon comes from the contamination from ambient environment. Red: Carbon profile in the SAMM-doped sample.

For the blank sample, SIMS data show a high concentration of carbon due to carbon physically absorbed on silicon surface (black line in Fig.D). While the silicon surface is being sputtered, the amount of carbon exponentially decays and reaches the SIMS carbon detection limit ($\sim 4 \times 10^{16} \text{ cm}^{-3}$) after the first 50nm silicon is removed. The SIMS carbon detection limit is confirmed by the technical expert at Evans Analytical Group where SIMS is performed. For the SAMM-doped sample, the carbon dopants first exponentially decay within the first ~40nm, reaching $\sim 5 \times 10^{17} \text{ cm}^{-3}$, and then slowly decays beyond the

depth of 200nm before going below the SIMS carbon detection limit. This clearly shows that there is carbon co-doped in the SAMM-doped sample. DLTS is probing defects in a region greater than 300nm below the surface where the concentration of carbon introduced by the SAM doping process is $\sim 10^{16} \text{ cm}^{-3}$ or a little lower.

We have added Fig.D and related discussions in the revised manuscript to support the conclusion drawn from the DLTS measurements..

Note: The surface carbon contamination is different for the two samples investigated. We cannot simply deduct the black line (first 50nm, in particular) from the red line to remove the carbon contaminants from the SAMM doped sample.

(17). In page 9, they should clarify how the rate windows related to the charge emission rate to obtain the peak on the DLTS signal.

Response: Thank the reviewer for reminding us of this point. But this requires describing in details the DLTS technique, which is not the purpose of this manuscript. Literature is full of details regarding this point. To accommodate the reviewer's request, it is more appropriate to add the related information in the supporting information. The following sentence has been added to the DLTS simulation section in the supporting information:

“The experimental rate window in the case of the boxcar method is used in the present work in which the transient is sampled at two instant t_1 and t_2 , with $t_2 > t_1$. A respond peak is detected when the trap's emission rate matches the rate window, which is the emission rate $e_0 = \ln(t_2/t_1)/(t_2 - t_1)$. For instance, when the rate window is 5 s^{-1} , t_1 is 0.122172 s and t_2 is 0.30543 s.”

(18). In the supplementary material, the DLTS simulation section, it is the only place where they mention the reverse and pulse bias condition, being -2 V / -0.5 V and -2 V / -1 V.... I have concerns with this values. The capacitance voltage measurements that they have not done and report, could tell us what is the depletion region at this voltage values.... they should report this to make sure that they are studying the region they claim to study, i.e., the first 200 nm from the sruface. Also, what was the time pulse?

Response: We have previously conducted CV measurements (see Fig. C above) but was not placed in the supporting information. Fig.C has now added into the revised supporting information and the related discussions have been updated in the revised manuscript.

We did not claim that DLTS is probing the first 200nm layer from the surface. There must be some misunderstanding, probably because we didn't make this clear in the original manuscript.

The phosphorus dopants are mostly located in the first 200nm layer according to the SIMS profile. As mentioned previously, conventional DLTS on Schottky device cannot allow for investigating the depletion region under 0V (roughly the top 300 nm in the bulk) so that the famous first 200 nm, included in this depletion region, are in principle inaccessible. What the DLTS actually probes is the area just beyond the depletion region at 0V (~300nm). The carbon profile in Fig.D clearly shows that carbon dopants ($\sim 10^{16}$ /cm³) from the SAMM doping process have reached this region. Therefore, it is not surprising that the DLTS has detected carbon-related defects formed by interstitial carbon bonding with the background phosphorus in the n-type substrate and possibly phosphorus dopants introduced by SAMM doping process. It is worth to note that DLTS is sensitive to concentrations down to 10^{11} cm⁻³.

The time pulse used in the present work is 1 μ s.

(19). Also, in their simulations Fig. 5, it looks to me they need more peaks to fit well the curves.

Response: Yes, we are aware of this. Of course we may always get closer to the ideal fit by adding new peaks contributing to the signal. But we do not think this helps to get a better insight into the physics. Again to justify this point we must go back to the detail of the DLTS treatment. The “theoretical peaks” have a given broadening or FWHM (Full Width at Half Maximum) and whenever the experimental peak is larger, it is an indication that the peak does not represent a single level but a distribution. The clue is to try to have a reasonable guess of the number of levels involved. Ideally the theoretical envelop must be exactly the same as the experimental spectrum. But we must be aware that other factors having nothing to do with what is happening in the band gap may bring some distortions, like the real barrier height and surface states which are difficult to assess with.

(20). In the DLTS section page 15, they mention the data was collected using Laplace DLTS software.... so are their DLTS curves, usual DLTS plots or Laplace DLTS plots?

Response: The DLTS curves are plotted in usual DLTS plots. In the experimental section DLTS on page 15, we updated the description to make it clear. “A conventional DLTS with boxcar mode are applied to get better resolution. Data was collected using Laplace DLTS software and plotted in usual DLTS plots.”

(21). In summary, I think the topic of research is interesting and suitable for nanocommunications. However the authors need to correct the electrical characterization part. Briefly:

1. Repeat van der Pauw measurements using the ohmic contacts mentioned here.

Response: As we responded in (3) previously, the metal contacts of our devices have no problem. Aluminum can form good Ohmic contact on n-type silicon, whereas gold can form good Schottky diode on n-Si, both without thermal annealing. This is consistent with the classical semiconductor device textbook. We also re-conducted the experiments to prove our statements (Fig.A and Fig.B).

2. Show IV curves that prove they made suitable ohmic contacts.

Response: please see IV curves in Fig.A for Al Ohmic contacts, and IV curves in Fig.B1, B2, and B3 for Ohmic contacts in van der Pauw measurements.

3. Redo the Schottky devices, using Ti for Schottky and annealed Au for the back ohmic contact. Remember to do FIRST the back contact and SECOND the Schottky contact. Clarify this clearly when you do it.

Response: Again, the way we fabricated our device contacts has no problem. We have revised the related part of the manuscript to clarify the fabrication process clearly.

4. Make IV and CV curves for the Schottky diode, at several temperatures.

Response: The related CV curves have been added into the supporting information. The IV curves for Schottky diode (Fig.3a and Fig.S5) at room temperature look beautiful. These IV curves will only look even better at low temperature, which is exactly what we saw during the experiments.

5. Specially analyse the CV curves to clearly determine the voltage parameters for the DLTS measurements such that they really study the first 200 nm layer from the surface where they claim the P-doped the sample.

Response: We have previously conducted the CV measurements. The CV data are now added in the supporting information and the related sections in the manuscript have also been updated. Please refer to our responses in (15) and (18) for detailed explanations.

For the 200nm layer, it is apparently a misunderstanding probably because we didn't make it clear in the original manuscript. In the revised manuscript, we have clearly stated this.

Conventional DLTS on Schottky device cannot allow for investigating the depletion region under 0V (roughly 300 nm) so that the famous first 200 nm, included in this depletion region, are in principle inaccessible. This is due to the doping considered in the present work. What the DLTS actually probes is the area just beyond the depletion region at 0V. The carbon profile in Fig.D clearly shows that a relatively high concentration ($10^{15}/\text{cm}^3 - 10^{16}/\text{cm}^3$) of carbon has diffused to a region beyond 300nm. The carbon dopants beyond 300nm bind more with the background phosphorus dopant in the n-type substrate and possibly less phosphorus dopants from the SAMM doping process, forming C_i-P_s pairs. DLTS is probing these defects which in the present work act as a marker for carbon.

6. Redo SIMS and try to see carbon profiling.

Response: Yes, we have re-done the carbon profiling using SIMS. The data are shown in Fig.D. Clearly, carbon diffuses far into the substrate, reaching beyond 300nm where carbon doping concentration is $\sim 10^{16} \text{ cm}^{-3}$ or a little less. This part of carbon eventually binds with the background phosphorus dopant in the n-type substrate or even phosphorus dopants from the SAMM doping process, forming C_i-P_s pairs. The DLTS signals are mostly coming from this part of carbon defects.

7. Correct all the other minor details that have been mentioned.

Response: We sincerely thank the reviewer for pointing out these minor issues, which is very helpful to improve the quality of our manuscript. We have corrected all these minor issues in the manuscript.

Good luck!

Response: Thank you! Have a nice day in the new year.

Reviewers' Comments:

Reviewer #2 (Remarks to the Author)

Authors have managed to improve their manuscript.

However, there are still some corrections to be done.

Nowhere in the manuscript it is mentioned the applied voltages that are used in the DLTS study. This is very important information. To access this information it is necessary to download the supplementary information. This information about the reverse biases should be added in the manuscript.

In the supplementary information it is found that the DLTS study consisted on two sets of applied voltage. Reverse bias was always -2 V, and the pulse bias was either -0.5 V or -1 V. It looks to the reviewer that with these applied voltages, it might be the case that the authors are not studying the layers close to the surface of the sample.

In order to be sure that the authors are effectively study the surface and the effect of P-doping, authors should include a band diagram of their device, for all these three applied biases, namely - 0.5 V, -1 V and -2 V.

They should do this integrating two times their figure S10 (which they should put it in the manuscript and not as supplementary material), from a depth deep enough, in order to achieve the builtin potential, plus the position of the fermi level relative to the conduction band offset plus the applied reverse bias. This under the assumption that it is the P concentration which yields the charge carrier concentration, which might not necessarily be true, as discuss next.

Also, when making the CV analysis, they should do the C^{-2} vs V analysis (show the plots and the linear fit) and report from the slope the charge carrier density and the built-in potential. This should also be included in the manuscript, not as supplementary material.

At first glance, according to their figure S10, it looks to the reviewer that the authors might not be studying a significant portion of the layer close to the substrate. As a rough estimation, it seems, from figure S10 that the depletion regions for reverse biases of -0.5, -1 V and -2 V are 390 nm, 485 nm, 650 nm.

If these values are correct, then, according to their Figure 2 a), they are not studying at all the P-doped region, and then the conclusions are wrong, as they their DLTS study is from such a depth, where the SAMM technique has practically not affected at all the Si layer.

So, briefly, after this discussion, that the is the reason why the band structure at those voltages are necessary, to have a clear idea of the depth of the depletion region at each of their reported applied voltages.

Once the authors have made these analysis, the reviewer will gladly review again the manuscript.

Reviewer #2 (Remarks to the Author):

Authors have managed to improve their manuscript.

However, there are still some corrections to be done.

1. Nowhere in the manuscript it is mentioned the applied voltages that are used in the DLTS study. This is very important information. To access this information it is necessary to download the supplementary information. This information about the reverse biases should be added in the manuscript.

Response: We thank the reviewer for reminding us this. We agree with the reviewer and have added this information in the caption of Fig.3 by clearly stating that the DLTS is recorded at the reversed-bias pulse from -2V to 0V. Accordingly, the depletion region edge sweeps from 663nm to 307nm (distance from the Au-Si interface). DLTS signals are from defects in this region.

2. In the supplementary information it is found that the DLTS study consisted on two sets of applied voltage. Reverse bias was always -2 V, and the pulse bias was either -0.5 V or -1 V. It looks to the reviewer that with these applied voltages, it might be the case that the authors are not studying the layers close to the surface of the sample.

In order to be sure that the authors are effectively study the surface and the effect of P-doping, authors should include a band diagram of their device, for all these three applied biases, namely -0.5 V, -1 V and -2 V.

Response: The range of pulse bias and the corresponding sweeping range of the depletion region are listed as following:

(0V, -2V) → (307nm, 663nm)

The DLTS data in Fig.3d in the manuscript is probing this region in which carbon binds more with P from SAMM process and less with the background P dopants in the n-Si substrate.

(-0.5V, -2V) → (426nm, 663nm)

The DLTS data in Fig.S10a of the updated SI is probing this region in which carbon binds dominantly with the background P dopants in the n-Si substrate.

(-1V, -2V) → (518nm, 663nm)

The DLTS data in Fig.S10b of the updated SI is probing this region in which carbon binds dominantly with the background P dopants in the n-Si substrate.

If we compare the DLTS signal magnitude, it is not hard to find that as the probing region moves away from the surface, the DLTS peaks continuously become smaller. This is consistent with the scenario in our case. When the depletion region moves to over 1 μm away from the surface, no DLTS signal will be detected because the carbon concentration will be lower than the DLTS detection limit (10^{11} cm^{-3}) (extending the carbon diffusion profile in Fig.6). This is exactly what we observed on the SAMM-doped intrinsic Si substrate in which the depletion extends a few micrometers away from the surface. Due to this reason, we repeated the experiments on n-Si substrate, i.e., to confine the depletion region close to the surface.

To clearly illustrate this, we plot the P and C profile and the band diagram at different bias for comparison. The figure is inserted in SI to help future readers have a better understanding.

3. They should do this integrating two times their figure S10 (which they should put it in the manuscript and not as supplementary material), from a depth deep enough, in order to achieve the built-in potential, plus the position of the fermi level relative to the conduction band offset plus the applied reverse bias. This under the assumption that it is the P concentration which yields the charge carrier concentration, which might not necessarily be true, as discuss next.

Also, when making the CV analysis, they should do the C^{-2} vs V analysis (show the plots and the linear fit) and report from the slope the charge carrier density and the built-in potential. This should also be included in the manuscript, not as supplementary material.

Response: We agree with the reviewer. We insert the $1/C^2$ vs V curves and the ionized charge profile extracted from the CV curves in Fig.3. Another paragraph is added to discuss these two panels.

Figure 3. (a) I-V curve of the Schottky diode made on the SAMM-doped sample with the inset schematically showing the diode structure. (b) Capacitance as a function of bias voltage in a form of $1/C^2$ vs V . (c) The charge carrier concentration at different depth, derived from (b). (d) Comparison of DLTS spectra of blank sample, control sample, and SAMM-doped sample with reversed-bias pulse from $-2V$ to $0V$, at the rate window of 200 s^{-1} . The inset shows the spectra at the range of 65 K to 85 K .

4. At first glance, according to their figure S10, it looks to the reviewer that the authors might not be studying a significant portion of the layer close to the substrate. As a rough estimation, it

seems, from figure S10 that the depletion regions for reverse biases of -0.5, -1 V and -2 V are 390 nm, 485 nm, 650 nm.

If these values are correct, then, according to their Figure 2 a), they are not studying at all the P-doped region, and then the conclusions are wrong, as they their DLTS study is from such a depth, where the SAMM technique has practically not affected at all the Si layer.

Response: These numbers are close to what we calculated as shown previously. But this does NOT mean our conclusions are wrong. It is mistaken to compare these numbers with the phosphorus profile in Fig.2a which is on the intrinsic Si substrate. The reviewer can recall that we repeated the experiments on the n-Si substrate with a background P doping concentration of $3 \times 10^{15} \text{cm}^{-3}$. We also employed SIMS to detect the P profile in this sample. The profile was previously placed in SI but now is added to Fig.6 in the manuscript along with the carbon profile.

In the response to Comment #2 above, we have made it clear what the DLTS is probing at the different pulse bias. In short, DLTS can detect carbon-related defects in our samples located less than $1 \mu\text{m}$ away from the surface. For all the DLTS measurements on SAMM-doped sample, the detection region is less than 663nm away from the surface. There is no concern that DLTS signals might be artificial.

The reviewer is right to state that both our DLTS and CV tools are probing the region just beyond the interesting one (SAMM region detected by SIMS). To be able to probe the SAMM area, we need either to increase the injecting pulse above 0V or keep the sequence (-2V to 0V) and use a high background doping, namely larger than $3 \times 10^{16} \text{cm}^{-3}$. In principle this value allows probing at least part of the decaying phosphorus profile due to SAMM. Solving Poisson equation asserts this. The problem is that in the present case none of these solutions is really helpful because it is difficult to implement. Pulsing the bias toward higher voltage saturates the capacitance meter because a large current is injected in the measuring bridge. This is why we restricted ourselves to 0V or even to negative values. The reason is precisely the high doping value near the surface. On the other hand using a background doping level above 10^{16}cm^{-3} , although in principle possible, turns out to be a nightmare as making Schottky diode on a highly doped surface remains very difficult. Again we would be facing a large surface leakage current which would prevent us from benefiting properly from CV data.

Fortunately, diffusion considerations save the issue in the sense that carbon contamination is extending much beyond the SAMM area and depending on the bias sequence the pair C_iP_s involves mainly P coming from SAMM or from the background doping.

When we magnify the profiles joined to the band diagrams shown in response to Comment #2, provided we take into account the 80% of the active dopant as determined by SIMS, we find exactly what is shown in Figure 3(c). So the results are coherent.

5. So, briefly, after this discussion, that the is the reason why the band structure at those voltages are necessary, to have a clear idea of the depth of the depletion region at each of their reported applied voltages.

Once the authors have made these analysis, the reviewer will gladly review again the manuscript.

Response: We agree with the reviewer and have plotted the band diagram at these voltages in the response to Comment #2.

Reviewers' Comments:

Reviewer #2:

Remarks to the Author:

The original intention of the manuscript was to study the incorporation of C and P into Si wafers due to the self-assembled molecular monolayer (SAMM) technique, and using the DLTS technique, get information about the defects these technique causes inside the Si wafer.

Now in this new version, they mention (Fig. 6) that they choosed a silicon wafer which already has a huge carbon doping from the beginning, almost 10^{21} cm^{-3} close to the surface, and it decreases to a non-despicable value of almost 10^{17} cm^{-3} ... this is a bad choice, because there is already a large amount of C already in the untreated samples.

Authors should have started with a Si wafer where there was not C and P at all inside, for example a Si wafer doped with N or As, preferably As, in order to have it n-type, but with no C and P at all.

It is difficult to believe that just from the atmosphere the Si wafer gets so huge amount of C doping considering that Si forms a native oxide on the surface that should hinder any introduction of impurities.

The reviewer is curious from where the authors got such C-doped Si wafer... did you order a Czochralsky (CZ) or Float Zone (FZ) wafer? what company sold you such a C-doped Si wafer? please provide this information to the reviewer.

The authors can not conclude what they say, if already from the beginning they have such a huge C doping in the Si wafer. Thus, author should repeat this experimental part, doing the following:

1. Get a Si wafer doped with N or As, preferably As, so it is n-type. Order a FZ Si wafer, to make sure the number of impurities are really small.
2. Get a SIMS profile of one piece of this wafer, as bought, and check that effectively it has no C or P from the beginning.
3. Do your SAMM process.
4. Again do SIMS profiling, this time to the SAMM-processed sample, confirming the incorporation of C and P into the wafer.
5. Do you Schottky and ohmic contacts as you have done before.
6. Do your DLTS study... BUT... the simplest way to effectively study the layer close to the surface, is to study the DLTS signal at zero voltage, and populate this region with a FORWARD BIAS... in this way you can be sure you are studying the layer close to the surface and the Schottky contact... see further details about this DLTS procedure in the following article: Nanotechnology Vol. 27 p. 075705 year 2016 and references 16 and 30 therein, you could cite some of them.

In this way you stop all kind of speculation of your DLTS spectra that you provide in your rebuttal letter, for example, you mention that for the 0V  -2 V C binds more with the P of the SAMM process and less with the background P, while the -0.5 V  -2 V and -1 V  -2 V C binds dominantly with the background P... how can the authors claim this if from the very beginning you have such a huge amount of C? and these sentences mention there is a background P dopant already, but in your Fig. 6, you just show ONE P profile in SAMM-doped n-type Si...not leaving clear if that P profile is similar to the P profile of the original Si wafer before the SAMM process... in other words, when redoing this experimental part, in the first SIMS profile, the one done to the

wafer before the SAMM process, show both the C and P profile (if any), and do the same, show both C and P profile of the Si wafer once the SAMM process is done.

In this way you can be sure that effectively your SAMM process is introducing C and P to a Si wafer which was originally empty of C and P.

Once authors have done this, the reviewer will gladly look again the manuscript.

Reviewers' comments:

Reviewer #2 (Remarks to the Author):

Q.1

The original intention of the manuscript was to study the incorporation of C and P into Si wafers due to the self-assembled molecular monolayer (SAMM) technique, and using the DLTS technique, get information about the defects these technique causes inside the Si wafer.

Now in this new version, they mention (Fig. 6) that they choosed a silicon wafer which already has a huge carbon doping from the beginning, almost 10^{21} cm^{-3} close to the surface, and it decreases to a non-despicable value of almost 10^{17} cm^{-3} ... this is a bad choice, because there is already a large amount of C already in the untreated samples.

Authors should have started with a Si wafer where there was not C and P at all inside, for example a Si wafer doped with N or As, preferably As, in order to have it n-type, but with no C and P at all.

It is difficult to believe that just from the atmosphere the Si wafer gets so huge amount of C doping considering that Si forms a native oxide on the surface that should hinder any introduction of impurities.

Response: In early 1980s when SIMS was first employed to study carbon impurities in solids, similar high concentrations of carbon profiles near the surfaces were often observed on many solids. It was also believed that the detected carbon impurities are located in the solid. This belief led to many bizarre conclusions. A famous researcher once wrote in his paper (Contrib. Mineral Petrol 76, 1981: 474-482) when he studied the carbon impurities in MgO using SIMS.: “ The

behavior of carbon in MgO is so strange in many respects that it sometimes appears almost unbelievable.”

Later on, many researchers uncovered the truth behind this. A very nice paper on this can be found here (Phys. Chem. Minerals 12, 1985:261-270) which is also attached at the end of this response letter. The conclusion is that the carbon impurities in SIMS come from the physical absorption of carbonates on the solid surfaces. The carbon impurities are not located inside the solid. Instead, they are always at the surface. When the substrate is sputtered during the SIMS, the carbonates absorbed on the surface will also be sputtered, creating a high concentration of carbon impurities in SIMS signals, although the substrate is free of carbon.

To support this claim, we also sent our samples to the headquarters of EAG Laboratories at New Jersey in USA to perform an advanced SIMS carbon profiling under high vacuum condition of about 3×10^{-11} Torr. Focused Cs^+ primary ion beam was applied for sputtering, which facilitate high yields of secondary ions of P and C. The trick to remove surface carbonates is to apply oxygen plasma cleaning on the surface right before SIMS sputtering. After this treatment, the SIMS carbon profile looks flat at $2\text{-}3 \times 10^{16} \text{ cm}^{-3}$ which is consistent with the certificated carbon concentration ($< 5 \times 10^{16} \text{ cm}^{-3}$) in our n-type Si wafers used in the experiments, as shown in Fig. R1.

Figure R1. Carbon in bare silicon wafer detected by SIMS. Black squares: carbon profile in P-doped silicon after ozone plasma cleaning. Red dots: carbon profile in P-doped silicon without plasma cleaning.

Q.2

The reviewer is curious from where the authors got such C-doped Si wafer... did you order a Czochralsky (CZ) or Float Zone (FZ) wafer? what company sold you such a C-doped Si wafer? please provide this information to the reviewer.

Response: The n-type silicon is CZ silicon from Resemi Company (<http://www.resemi.com/>). According to the product certification they offered us, the resistivity is 1~3 $\Omega \cdot \text{cm}$, corresponding to the phosphorus concentration of $\sim 3 \times 10^{15} / \text{cm}^3$. The carbon concentration is $< 5 \times 10^{16} / \text{cm}^3$.

Q.3

The authors can not conclude what they say, if already from the beginning they have such a huge C doping in the Si wafer.

Response: If there is indeed a huge carbon concentration inside the wafer, it is true that we cannot reach our conclusion. But as we show in the response to Q.1 above, the carbon concentration is as low as $2 \times 10^{16} \text{cm}^{-3}$ in our n-type Si wafer. Therefore, it is still possible for us to conclude that there is a high concentration of C_i-P_s defects in the SAMM-doped region. In particular, following the suggestions that the referee provided later in this letter, we conducted the following experiments on our samples.

First, we profiled the P and C concentration in the SAMM-doped sample and the blank sample, as shown in Fig.2. Clearly, the SAMM doping process introduced a significant portion of C and P into the substrate (within the first 300nm).

Figure R2. SIMS profiles for C and P in the blank sample and SAMM-doped sample.

Second, to show clearly that carbon defects bind with P dopants introduced by the SAMM doping process, we tuned the bias voltages from -2V to 0V and injection pulses from -1V to 0.2V, pushing the DLTS to probe from bulk to near the surface. We plotted the C and P profiles along with the probing region for comparison to show the C and P concentration in each region. For each probing region, we also put the DLTS data side by side with the probing region (gray area), as shown in Fig. R3 below. Clearly, the bias pulse from 0V to +0.2V is probing the defects formed by C and P introduced by the SAMM-doping process. For more detailed discussions, please kindly refer to the updated manuscript and supplementary information.

Figure R3. P and C depth profiles by SIMS compared with ionized charge profile derived from CV (a). Silvaco simulation on band structure at 300K with bias voltage of 0V (b), -0.2V (d), -2V (f) and -2V (h). Probe regions are shaded in grey with different pulses from 0V ~ 0.2V (b), -0.2V ~ 0.2V (d), -2V ~ 0V (f) and from -2V ~ -1V (h). DLTS simulations on spectra of SAMM-doped silicon with pulses from 0V ~ 0.2V (c), -0.2V ~ 0.2V (e), -2V ~ 0V (g) and from -2V ~ -1V (i). The rate windows of DLTS spectra are 200 s⁻¹.

Q.4

Thus, author should repeat this experimental part, doing the following:

1. Get a Si wafer doped with N or As, preferably As, so it is n-type. Order a FZ Si wafer, to make sure the number of impurities are really small.
2. Get a SIMS profile of one piece of this wafer, as bought, and check that effectively it has no C or P from the beginning.
3. Do your SAMM process.

4. Again do SIMS profiling, this time to the SAMM-processed sample, confirming the incorporation of C and P into the wafer.

5. Do you Schottky and ohmic contacts as you have done before.

6. Do your DLTS study... BUT... the simplest way to effectively study the layer close to the surface, is to study the DLTS signal at zero voltage, and populate this region with a FORWARD BIAS... in this way you can be sure you are studying the layer close to the surface and the Schottky contact... see further details about this DLTS procedure in the following article: Nanotechnology Vol. 27 p. 075705 year 2016 and references 16 and 30 therein, you could cite some of them.

In this way you stop all kind of speculation of your DLTS spectra that you provide in your rebuttal letter, for example, you mention that for the 0V  -2 V C binds more with the P of the SAMM process and less with the background P, while the -0.5 V  -2 V and -1 V  -2 V C binds dominantly with the background P... how can the authors claim this if from the very beginning you have such a huge amount of C? and these sentences mention there is a background P dopant already, but in your Fig. 6, you just show ONE P profile in SAMM-doped n-type Si....not leaving clear if that P profile is similar to the P profile of the original Si wafer before the SAMM process... in other words, when redoing this experimental part, in the first SIMS profile, the one done to the wafer before the SAMM process, show both the C and P profile (if any), and do the same, show both C and P profile of the Si wafer once the SAMM process is done.

In this way you can be sure that effectively your SAMM process is introducing C and P to a Si wafer which was originally empty of C and P.

Once authors have done this, the reviewer will gladly look again the manuscript.

Response: Although we think our conclusion is supported after we performed the experiments and collected the CV and DLTS data shown above, we still repeated the experiments on the As-doped substrate following the referee's suggestions from Step 1 to Step 6.

First, the C and P concentration were profiled by SIMS for the SAMM-doped sample and the n-type blank sample (As-doped $\sim 1.4 \times 10^{15}/\text{cm}^3$), as shown in Fig. R4 below. Note that the detection limit of SIMS for C and P are both around 10^{16} cm^{-3} .

Second, we made Schottky contacts to both the SAMM-doped sample and the blank sample. The IV curves are shown in Fig. R5a. From the CV curves, we extracted the ionized charge profile for the SAMM-doped sample (red dots) and the blank sample (black dots) in Fig. R5b. Indeed, the As doping concentration in the bulk is $\sim 1.4 \times 10^{15}/\text{cm}^3$ based on the ionized charge profile. The P profile by SIMS is plotted together for comparison. The ionized charge profile extracted from CV and the P profile by SIMS are largely consistent.

We also tuned the bias pulse from negative to mostly positive. Strong DLTS signals are observed when the Schottky junction is biased at -0.5V but pulsed to 0.2V, as shown in Fig. R5c, in which the P dopants introduced by the SAMM doping process are dominant (Fig.R5b).

We changed the rate window from 5 s^{-1} to 100 s^{-1} and intended to extract the defect energy levels as we did on the phosphorus-doped sample. However, the DLTS signals are in form of broad envelopes (Fig.R5d). It is difficult to identify the multiple peaks buried in the envelopes. From [MRS Proceedings, 1989, Cambridge Univ Press: p 295], interstitial carbon can bond with group V elements. Therefore, the DLTS spectra in Fig.R5 likely contain information from $\text{C}_i\text{-P}_s$ and $\text{C}_i\text{-As}_s$ defects, which makes our analysis of the defect energy extremely difficult. In particular, we have no clue of the $\text{C}_i\text{-As}_s$ defect levels since no one has ever studied the $\text{C}_i\text{-As}_s$ defects as far as we know. But clearly some sort of $\text{C}_i\text{-As}_s$ defects are formed in the SAMM-doped sample because strong DLTS signals are observed at 263 K in the spectrum but not in the DLTS spectra for $\text{C}_i\text{-P}_s$ defects presented in the manuscript.

Due to this reason, we decided not to use the data on the As-doped substrates to replace the data on the P-doped substrates in the original manuscript. In particular, our data on the original substrates are complete and convincing (Fig.R3).

Reviewers' Comments:

Reviewer #2:

Remarks to the Author:

Authors have managed to improve their manuscript.

However, there are several inaccuracies in the electrical part that should be address first in order to make this article publishable, but specially, scientifically valuable.

In their Fig. 3, and citing their own caption, authors write "Capacitance as a function of bias voltage in form ..." however the units are $(\text{cm}^2/\text{F})^2$... i.e., either the units are wrong or it is not $1/C^2$... authors should correct this.

Also, in all their DLTS graphic, they add as units of DLTS (μF)... there is something wrong on this, the DLTS signal is the change of capacitance per capacitance, thus, it has no units, check carefully the paper by D. V. Lang where he introduces the DLTS technique,... there is something wrong in their DLTS units, as it should be unitless.

Regarding deconvolution of their DLTS spectra, it could be done better, as the envelope does not reproduce fully the spectra, namely, they should include peaks (and explain them):

1. in Fig. 4e, between 1 and 2, 2 and 3, and 3 and 4.
2. in Fig. 4f, between 1 and 2, 2 and 3, and 3 and 4.
3. in Fig. 5c, around 200 K.
4. in Fig. 5e, around 200 K.
5. In Fig. 5g, around 175 K, 200 K and 230 K.
6. There is enough space in Fig. 5 i) to plot with more detail the graphic, it is almost impossible to see it well, please plot it better.

In Fig. S4 caption, word "concentration" is wrong.

In Fig. S6, y-axis should be in scientific notation, not engineer notation.

In Fig. S7, why didnt the authors showed the CV measurement for $V > 0$ V for control sample? they should plot it.

In Fig. 5 a-h) and S9 b-e), they should indicate where is the Fermi level, it would be convenient to set it at zero. Also, no y-axis is available, so readers can not know the values of energy.

Also, nowhere authors report the built-in potential of the Schottky diode, that is obtained from the CV analysis... they should report it, it is important, as it should be different from 300 K and 50 K, according to their Fig. 3 b).

Once authors have done these corrections, the reviewer will gladly review the manuscript again.

Reviewer #2 (Remarks to the Author):

Authors have managed to improve their manuscript.

However, there are several inaccuracies in the electrical part that should be addressed first in order to make this article publishable, but especially, scientifically valuable.

(1). In their Fig. 3, and citing their own caption, authors write "Capacitance as a function of bias voltage in form ..." however the units are $(\text{cm}^2/\text{F})^2$... i.e., either the units are wrong or it is not $1/C^2$... authors should correct this.

Response: We thank the reviewer for bringing this to our attention. In previous version of the figure, we used capacitance per square centimeter to plot the correlation between capacitance and bias voltage, therefore the unit of $1/C^2$ should be cm^2/F^2 .

To have a clear presentation, we have changed the y-axis parameter to total capacitance as seen in the figure below, thus the unit is $1/\text{F}^2$. Figure 3 in the manuscript has also been updated.

Figure 1. Capacitance as a function of bias voltage in form of $1/C^2$ vs V .

(2). Also, in all their DLTS graphic, they add as units of DLTS (pF)... there is something wrong on this, the DLTS signal is the change of capacitance per capacitance, thus, it has no units, check carefully the paper by D. V. Lang where he introduces the DLTS technique,... there is something wrong in their DLTS units, as it should be unitless.

Response: Indeed, DLTS signals are more often plotted in relative capacitance ($\Delta C/C$) with arbitrary unit. But there is nothing wrong to plot in absolute capacitance with unit of pF. In fact, D.V. Lang did so as shown in Fig.9 in his landmark paper in 1974 (J. Appl. Phys. 45, 3023 (1974)). Of course, if the reviewer insists that we plot in relative capacitance with no unit, we can do that too.

The units of DLTS signals depend on the transient signals that the DLTS system collects. As far as we know, the transient signal can be transient capacitance, transient voltage, transient current or relative capacitance $\Delta C/C$. Accordingly the units are pF^1 ,

², mV³, μ A⁴, and arbitrary unit⁵. Our DLTS system is a boxcar DLTS, we detect capacitance transient like Fig.2 in (J. Appl. Phys. 62, 576-581 (1987)); Fig. 5 in (Rev. Sci. Instrum. 51, 2038-2042(1980)); Fig.9 in D. V. Lang's (J. Appl. Phys. 45, 3023 (1974)).

We are both right. But we prefer to keep pF there, to give more information.

(3). Regarding deconvolution of their DLTS spectra, it could be done better, as the envelope does not reproduce fully the spectra, namely, they should include peaks (and explain them):

1. in Fig. 4e, between 1 and 2, 2 and 3, and 3 and 4.

2. in Fig. 4f, between 1 and 2, 2 and 3, and 3 and 4.

3. in Fig. 5c, around 200 K.

4. in Fig. 5e, around 200 K.

5. In Fig. 5g, around 175 K, 200 K and 230 K.

Response to #1-5: We thank the reviewer for pointing this out. We have added more peaks to the simulation to yield better fitting results in the related figures (see figures below), as the reviewer suggested. Our concern is that we are not sure about the origin of these peaks. It could be related to surface states, other contamination-related defects such as nitrogen⁶ and atomic disorder⁷.

Since these peaks are not the main feature of the DLTS spectra, in order not to distract the readers' attention, we decide to put these figures in the supplementary information and keep the original figures as they are in the manuscript. At the end of the first paragraph on Page 14, we add the following sentence to point this out and direct the readers' attention to these figures in supplementary information:

“It is worth pointing out that the DLTS envelope by simulations does not match the experimental results perfectly. Some other peaks clearly exist, which may originate from surface states, nitrogen contaminants³³ or atomic disorder³⁴. The full deconvolution of the DLTS spectra can be found in Supplementary Figures 9-11 and Supplementary Table 3”

Figure 2. DLTS simulation on spectra (rate window of 5 s^{-1}) of the control sample (a) and the SAMM-doped Si (b). Note that the DLTS signals in (a) are much smaller in amplitude than those in (b).

Figure 3. (a) P and C depth profiles by SIMS compared with ionized charge profile derived from CV. Silvaco simulation on band structure at 300K with bias voltage of 0V (b), -0.2V (d), -2V (f) and -2V (h). Probing regions are shaded in grey with different pulses from 0V ~ 0.2V (b), -0.2V ~ 0.2V (d), -2V ~ 0V (f) and from -2V ~ -1V (h). DLTS simulations on spectra of SAMM-doped silicon with pulses from 0V ~ 0.2V (c), -0.2V ~ 0.2V (e), -2V ~ 0V (g) and from -2V ~ -1V (i). The rate windows of

DLTS spectra are 200 s^{-1} .

Table 1. List of energy levels derived from the extra fitting peaks in DLTS simulation. Extra peaks, peak a, b, c and d are in bold

	Bias pulse	Peak 0	Peak 1	Peak a	Peak 2	Peak b	Peak c	Peak 3	Peak d	Peak 4	Peak 5	Peak 6
Control sample	-2V ~ 0V	102	254	290	319	--	350	378	410	390	467	480
SAMM -doped sample	-2V ~ 0V	--	252	290	319	338	358	380	398	390	469	480
	-2V ~ -1V	--	260	290	319	338	358	380	--	390	467	480
	-0.2V ~ 0.2V	--	260	290	319	339	--	380	--	390	467	--
	0V ~ 0.2V	--	260	290	319	338	358	--	--	395	467	--

6. There is enough space in Fig. 5 i) to plot with more detail the graphic, it is almost impossible to see it well, please plot it better.

Response to #6: We thank the reviewer for this comment. We plotted in this way to compare the magnitude of DLTS peaks by using the same y-axis scale. The comparison can readily show that there are more defects near surface (in Figure 6c and e) than away from the surface (in Figure 6i). But we also agree with the reviewer that an enlarged peak should be plotted. For this reason, we put the enlarged image as Supplementary Figure 11. The decomposed peaks are summarized in Supplementary Table 3.

In the caption of Fig.6 in the manuscript, we add the following sentences to clarify this point:

“Note that panel c, e, g and i has the same y axis scale for better comparison. A close-up figure for panel i to show the fitting envelop can found in Supplementary Figure 11.”

Figure 4. A close-up of DLTS spectrum. (a) Deconvolution of main peaks in the DLTS spectrum. (b) Full deconvolution of the DLTS spectrum.

(4). In Fig. S4 caption, word "concentration" is wrong.

Response: We thank the reviewer for pointing this out. It has been corrected.

(5). In Fig. S6, y-axis should be in scientific notation, not engineer notation.

Response: We agree with the reviewer. Fig. S6 has been modified as seen below.

Figure S6. I-V curve of the Schottky diode on control sample

(6). In Fig. S7, why didnt the authors showed the CV measurement for $V > 0$ V for control sample? they should plot it.

Response: We agree with the reviewer's comment. At low temperatures, we can scan the voltage up to more positive value in CV measurement. We have replaced the CV curve of control sample at 300 K with the one at 50 K. Please also refer to the figure below for the CV curves of the control sample.

Figure 6. C-V curve of the Schottky diode on control sample and SAMM-doped sample

(7). In Fig. 5 a-h) and S9 b-e), they should indicate where is the Fermi level, it would

be convenient to set it at zero. Also, no y-axis is available, so readers can not know the values of energy.

Response: We agree with the reviewer. Fermi levels and y-axis information have been added to the updated figures in the manuscript and SI.

Figure 7. P and C depth profiles by SIMS compared with ionized charge profile derived from CV (a). Silvaco simulation on band structure at 300K (black line) and 50K (red line) with bias voltage of 0V (b), -0.2V (c), -2V (d) and -2V (e). Conduction band minimum and valence band maximum at 300K are draw in black solid line, Fermi level at 300K is in black dash line. Conduction band minimum and valence band maximum at 50K are draw in red solid line, Fermi level at 50K is in red dash-dot line.

(8). Also, nowhere authors report the built-in potential of the Schottky diode, that is obtained from the CV analysis... they should report it, it is important, as it should be different from 300 K and 50 K, according to their Fig. 3 b).

Response: We thank the reviewer for pointing this out. For the control sample that is uniformly doped with P dopants ($\sim 10^{15} \text{ cm}^{-3}$), we have obtained the built-in potential of the Schottky diode which is 0.57 eV at 300 K and 0.76 eV at 50K (see Fig.8 below).

As for the SAMM doped sample, it is difficult to extract the built-in potential because the P dopants are highly non-uniform, which results in a nonlinear correlation between $1/C^2$ and bias voltage (Fig.9b below). It is not accurate to derive the built-in potential directly from the C-V analysis of the SAMM doped sample.

In the middle of the second paragraph on Page 8, we have inserted the following sentences to clarify this:

“As expected, this built-in potential increases to 0.76V as the temperature is lowered to 50K (see Supplementary Figure 7). For the SAMM doped sample, the dependence of $1/C^2$ on dc voltage bias is nonlinear due to the highly non-uniform distribution of P dopants introduced by the SAMM doping process. This nonlinearity makes it unreliable to extract the built-in potential.”

Figure 8. Comparison of C-V of control sample at 300K and 50K. built-in potential increased from 0.57 eV to 0.76 eV as the temperature decreased from 300K to 50K.

Figure 9. C-V of control sample and SAMM-doped sample. Black arrows show why built-in potential derived from CV of SAMM-doped sample is not accurate.

Once authors have done these corrections, the reviewer will gladly review the manuscript again.

References

1. Pearce, N. O.; Hamilton, B.; Peaker, A. R.; Craven, R. A. *Journal of Applied Physics* **1987**, 62, (2), 576-581.
2. Pons, D.; Mooney, P. M.; Bourgoïn, J. C. *Journal of Applied Physics* **1980**, 51, (4), 2038-2042.
3. McLarty, P. K.; Ioannou, D. E.; Colinge, J.-P. *IEEE electron device letters* **1988**, 9, (10), 545-547.
4. Ayres, J. *Microelectronic Engineering* **1992**, 19, (1-4), 179-182.
5. Micocci, G.; Rizzo, A.; Siciliano, P.; Tepore, A. *physica status solidi (a)* **1989**, 114, (1), 253-257.
6. Tokumaru, Y.; Okushi, H.; Masui, T.; Abe, T. *Japanese Journal of Applied Physics* **1982**, 21, (7A), L443-L444.
7. Mesli, A.; Kringhøj, P.; Nylandsted Larsen, A. *Physical Review B* **1997**, 56, (20), 13202-13217.